# Audiovisual task switching rapidly modulates sound encoding in mouse auditory cortex

**Ryan J Morrill**[1,2,3], **James Bigelow**[1,3], **Jefferson DeKloe**[1,3], **Andrea R Hasenstaub**[1,2,3]*

[1]Coleman Memorial Laboratory, University of California, San Francisco, San Francisco, United States; [2]Neuroscience Graduate Program, University of California, San Francisco, San Francisco, United States; [3]Department of Otolaryngology–Head and Neck Surgery, University of California, San Francisco, San Francisco, United States

**Abstract** In everyday behavior, sensory systems are in constant competition for attentional resources, but the cellular and circuit-level mechanisms of modality-selective attention remain largely uninvestigated. We conducted translaminar recordings in mouse auditory cortex (AC) during an audiovisual (AV) attention shifting task. Attending to sound elements in an AV stream reduced both pre-stimulus and stimulus-evoked spiking activity, primarily in deep-layer neurons and neurons without spectrotemporal tuning. Despite reduced spiking, stimulus decoder accuracy was preserved, suggesting improved sound encoding efficiency. Similarly, task-irrelevant mapping stimuli during inter-trial intervals evoked fewer spikes without impairing stimulus encoding, indicating that attentional modulation generalized beyond training stimuli. Importantly, spiking reductions predicted trial-to-trial behavioral accuracy during auditory attention, but not visual attention. Together, these findings suggest auditory attention facilitates sound discrimination by filtering sound-irrelevant background activity in AC, and that the deepest cortical layers serve as a hub for integrating extramodal contextual information.

*For correspondence:
andrea.hasenstaub@ucsf.edu

**Competing interest:** The authors declare that no competing interests exist.

## Editor's evaluation

This is an important paper that is methodologically compelling, harnessing a complex behavioral task for modality-specific control of attention to provide new evidence that directed auditory attention produces a global decrease in auditory cortex firing rates without a loss of stimulus-related information. These findings build on previous results showing that task engagement or locomotion down regulates activity in auditory cortex. The manuscript is comprehensive and well-illustrated. It provides highly detailed analysis of the cortical activity modulations during attentional switching that will be valuable to others within and beyond the field of hearing research.

## Introduction

Information from one or another sensory pathway may become differentially relevant due to environmental changes. The brain must therefore continuously assign limited attentional resources to processing simultaneous information streams from each sensory modality. For example, hearing a siren while listening to music in the car might prompt an attentional shift away from the auditory stream, toward a visual search for emergency vehicles. On the other hand, a similar shift away from the music is unlikely while listening at home. In these cases, contextual cues support allocating attention to either the auditory domain or the visual domain, and the perceptual experience of the music is

qualitatively different. How might sensory cortex differentially encode stimuli from an attended versus filtered modality?

Attentional selection operates cooperatively at many levels of sensory processing. Most effort has been devoted to understanding the neural mechanisms of feature-selective attention within a single modality (*Desimone and Duncan, 1995*; *Fritz et al., 2007*). A major focus of this work has been characterizing transformations of stimulus representations in sensory cortical areas, due to their pivotal position between ascending sensory pathways and behavioral networks implementing top-down control (*Lamme et al., 1998*; *Sutter and Shamma, 2011*). These studies, largely from the visual domain, have shown that attention to a stimulus feature or space will often increase stimulus-evoked spiking responses and reduce thresholds for eliciting a response; likewise, responses to unattended stimuli are often decreased (*Reynolds and Chelazzi, 2004*). On the other hand, fewer studies have examined how modality-selective attention affects encoding in sensory cortex. This mode of attention highlights behaviorally relevant sensory streams while filtering less relevant ones. Human fMRI studies have reported differential activation patterns in auditory and visual cortex (AC, VC) reflecting the attended modality (*Johnson and Zatorre, 2005*; *Petkov et al., 2004*; *Shomstein and Yantis, 2004*; *Woodruff et al., 1996*). Extending these findings, studies in primate AC and VC have reported entrainment local field potential (LFP) oscillations by modality-selective attention, which serves to modulate excitability and sharpen feature tuning within sensory cortex corresponding to the attended modality (*Hocherman et al., 1976*; *Lakatos et al., 2009*; *Lakatos et al., 2008*; *O'Connell et al., 2014*). Several findings suggest that these influences may differ among cortical layers and between inhibitory and excitatory neurons (*Lakatos et al., 2016*; *O'Connell et al., 2014*).

Nevertheless, there are many open questions about the influence of modality-specific attention on stimulus encoding in sensory cortex. Importantly, potential interplay between ongoing activity and evoked responses during attentional selection, as well as their consequences for information and encoding efficiency, has not been examined. The degree to which influences of modality-specific attention may generalize beyond training stimuli has yet to be elucidated. Finally, how these influences may be differentially expressed in cell subpopulations defined by cortical depth or inhibitory/excitatory cell type similarly remains unknown.

In the present study, we addressed these open questions by examining single neuron activity and sensory responses in mouse AC during an audiovisual (AV) attention shifting task. AC integrates ascending auditory information with diverse input from frontal, cingulate, striatal, and non-auditory sensory areas to rapidly alter sensory processing in response to changing behavioral demands (*Budinger et al., 2008*; *Budinger and Scheich, 2009*; *Park et al., 2015*; *Rodgers and DeWeese, 2014*; *Winkowski et al., 2013*). To isolate the influence of modality-selective attentional modulation, we compared responses to identical compound auditory-visual stimuli under different cued contexts requiring attention to the auditory or visual elements, thus holding constant other task-related variables such as arousal, attention, reward expectation, and motor activity (*Saderi et al., 2021*). Because spike rate and information changes are dissociable (*Bigelow et al., 2019*; *Phillips and Hasenstaub, 2016*), we quantified both evoked spike rates and the mutual information (MI) between responses and stimuli. We also examined the generality of modality-specific attention by examining responses to task-irrelevant sounds presented between trials. Finally, we used translaminar probes and spike waveform morphology classification to capture possible attention-related differences in neurons among cortical layers and between putative inhibitory and excitatory cell classes.

## Results

### AV rule-switching in mice

We trained mice to perform an AV rule-switching task, in which they made decisions using auditory stimuli while ignoring simultaneously presented visual stimuli or vice versa. Trial presentation was self-paced in a virtual foraging environment wherein a visual track was advanced by forward locomotion on a spherical treadmill (*Figure 1A*). A task-irrelevant random double sweep (RDS) sound was presented during inter-trial intervals (ITIs) for mapping auditory receptive fields in each attentional state (*Figure 1B*). Decision stimuli were presented after variable track length, consisting of 1 s auditory tone clouds (TCs; centered at 8 or 17 kHz) and/or visual drifting gratings (horizontal or vertical orientation; *Figure 1C*). One of the decision stimuli for each modality was a rewarded target ($A_R$, $V_R$)

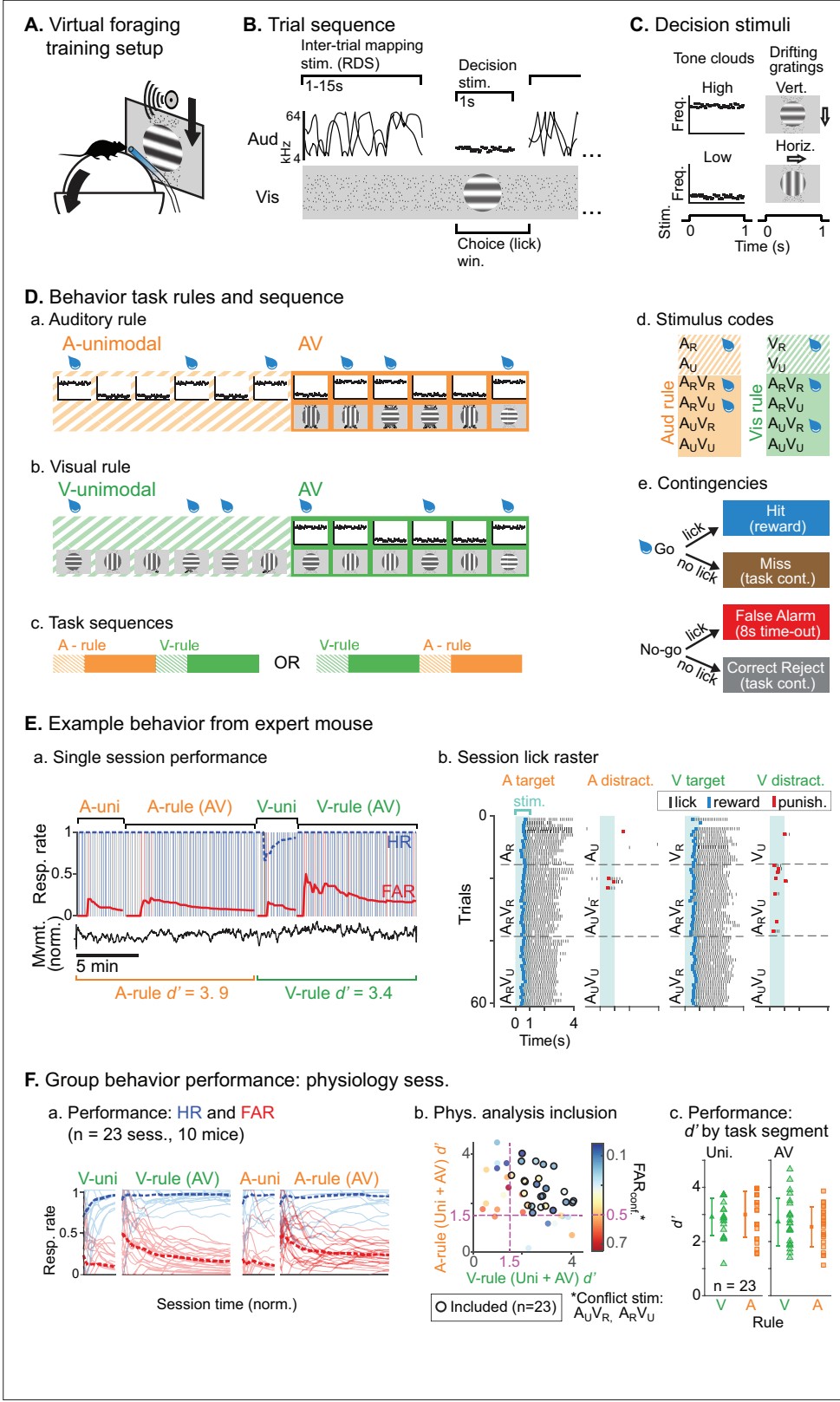

**Figure 1.** Audiovisual rule-switching in mice. (**A**) Virtual foraging environment: a head-fixed mouse runs on a floating spherical treadmill. Locomotion measured by treadmill movement controls auditory and visual stimulus presentation. A water spout in front of the mouse provides rewards. A lickometer records licks, which determines reward or punishment. (**B**) Trial sequence: during inter-trial intervals, a track of moving dots provides visual

*Figure 1 continued on next page*

*Figure 1 continued*

feedback for task progression while a random double sweep (RDS) auditory mapping stimulus is presented. Decision stimuli, either unimodal (auditory, A; visual, V) or bimodal (AV), are presented for 1 s. The choice window begins at decision stimulus onset, but trials with early licks (<0.3 s post-stimulus onset) are removed from subsequent analysis. (**C**) Decision stimuli are tone clouds (TCs; 8 or 17 kHz centered) or drifting gratings (horizontal or vertical orientation). Each mouse is trained to lick for one auditory stimulus and one visual stimulus. Target/ distractor stimulus identities were counterbalanced across mice for A- and V-rules. (**D**) Task sequences, attention cueing, and reward contingencies. (**a–b**) Behavioral sessions begin with a unimodal block, which cue the rule for the subsequent AV block. Water drops represent target stimuli, when mice have an opportunity for reward. (**c**) Each session used one of two possible task sequences. (**d**) Stimulus codes, for reference. (**e**) Contingencies for water reward, timeout punishment, or task continuation. (**E**) Example behavior session. (**a**) Hit rate (HR) and false alarm rate (FAR) across task blocks; trials and outcomes indicated by colored background bars. Mouse locomotion is shown below. (**b**) Stimulus onset-aligned lick rasters for example session, organized by rule and target/distractor. Note that errors are typically false alarms on trials with 'conflict' stimuli: $A_UV_R$ in A-rule or $A_RV_U$ in V-rule. (**F**) Performance for all sessions included in subsequent physiology analysis. (**a**) HR and FAR for all sessions organized by rule block; dashed lines indicate means. (**b**) Performance metrics, showing dual inclusion filters: 1. sensitivity index $d'$ performance index >1.5 for both A-rule and V-rule and 2. $FAR_{conf}$ <0.5 for conflict stimuli, as a critical test of modality-selective attention. (**c**) $d'$ is similar across task rules in unimodal and AV segments.

and the other an unrewarded distractor ($A_U$, $V_U$). Lick responses following targets (hits) and distractors (false alarms [FAs]) produced water rewards and dark timeouts, respectively. Withholding licks for targets (misses) or distractors (correct rejects [CRs]) advanced the next trial. Each session began with a block of unimodal decision stimuli, which cued the attended modality of a subsequent AV block (*Figure 1D*). A second unimodal block from the other modality was then presented, cueing the rule for a final AV block. Decision stimuli had identical physical properties but different behavioral significance between rules (e.g., licks following $A_RV_U$ were rewarded in A-rule but punished in V-rule). Targets and distractor stimuli remained constant throughout training for each mouse and were approximately counterbalanced across animals. Block sequences (A-rule then V-rule, or vice versa) were also counterbalanced across sessions (*Figure 1D*.c).

We used two approaches to ensure that animals were engaged during both task rules. First, we restricted analysis to sessions in which discrimination was well above chance ($d'$>1.5) for both rules, and for which FA rates were below 0.5 for the stimuli with reward valences that conflicted across rules ($A_UV_R$ in the A-rule, $A_RV_U$ in the V-rule; *Figure 1F*). Second, for a subset of sessions (*n*=14 sessions, 5 mice) we measured pupil size, a well-established correlate of arousal and behavioral performance (*Bradley et al., 2008*; *McGinley et al., 2015*; *Reimer et al., 2014*). We used a computer vision algorithm to automate measurement of pupil size (pupil diameter/eye diameter) for each frame acquired by a CCD video camera (*Figure 2A*.a). To isolate pupil fluctuations reflecting general arousal, pupil size was measured during an ITI window designed to avoid pupil responses to decision stimulus onset, dark timeouts, and decreased locomotion events following reward administration (*Figure 2*.A.b–c, *Figure 2B*). Previous studies have reported that pupil size increases with task difficulty and engagement in humans, non-human primates, and rodents (*Hess and Polt, 1964*; *Kawaguchi et al., 2018*; *Schriver et al., 2018*). We reasoned that comparison of pupil size across the rules would allow us to establish whether task demands differed between the rules. No difference in pupil size was observed between rules during bimodal blocks (*Figure 2C*; A-rule bimodal: 0.29±0.05 norm. diameter ± SD, V-rule bimodal: 0.30±0.05; *Z*=−1.0, p=0.30, paired Wilcoxon signed-rank [WSR], Benjamini-Hochberg false discovery rate [FDR]-adjusted p-values). As expected, pupil diameters were significantly larger during the bimodal portion of the task, when visual stimuli were present and the task had increased in difficulty, compared to the auditory-only unimodal portion of the task (*Figure 2—figure supplement 1*; A-rule unimodal: 0.28±0.04, A-rule bimodal: 0.29±0.05; *Z*=−2.6, p=0.009). A trend toward smaller pupil size in the unimodal visual rule compared to bimodal rule was also noted, but the difference did not reach significance after multiple comparisons correction (V-rule unimodal: 0.29±0.04, V-rule bimodal: 0.30±0.05; *Z*=−2.0, p=0.062). Because pupil size also closely tracks locomotion (*Figure 2A*.b–c; *McGinley et al., 2015*), we examined locomotion speed during the same ITI window (*Figure 2D*). Differences in locomotion speed were also not observed between rules (all p≥0.623; all |*Z*|≤1.29, paired WSR). Arousal and motor activity were thus comparable between rules, suggesting that differences in neuronal activity may be attributable to modality-selective attention.

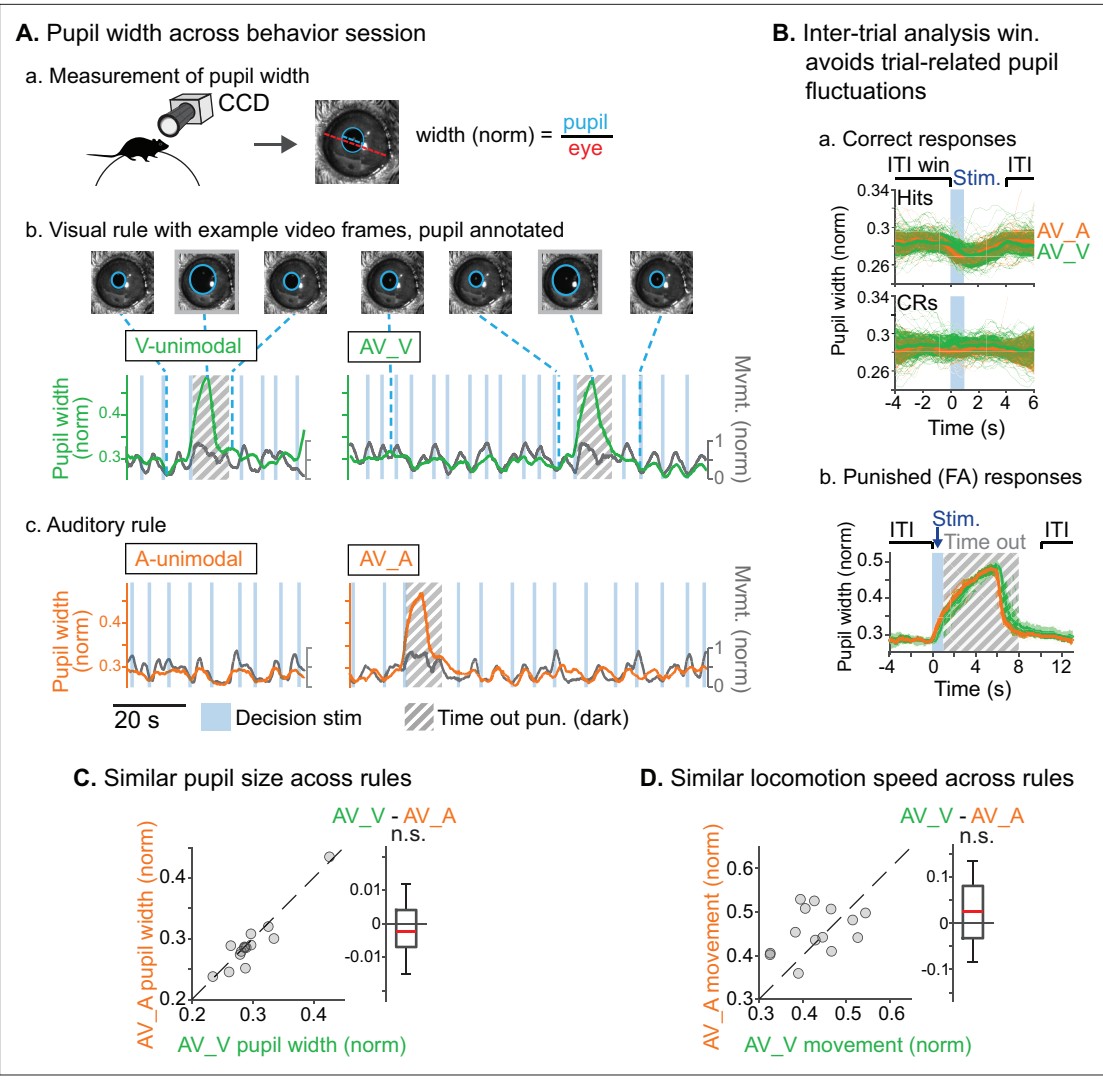

**Figure 2.** Similar levels of arousal and movement during auditory and visual attention. (**A**) Pupil size measurement. (**a**) Left eye pupil recorded via CCD camera during the task. Pupil circumference (light blue) is tracked using automated video analysis; size is measured as pupil diameter over visible eye diameter. (**b**) Example pupil video recorded during visual rule. Upper: annotated sample frames from times indicated by blue dashed lines. Lower: pupil width (green) and locomotion (gray) traces, with target stimuli and timeout punishments indicated. Large fluctuations of pupil size occur during timeouts due to drop in light level (hashed gray background). (**c**) Auditory rule from the same session. (**B**) Pupil size is measured during an inter-trial interval (ITI) window selected to capture engagement and arousal levels during each block and minimize influence from trial-related events such as rewards and timeouts. (**a**) Pupil size decreases during hit trials due to reward administration. Correct reject trials (CRs; bottom) show no such decrease in running speed. (**b**) Pupil size increases during timeout punishment when the recording chamber goes dark; ITI pupil size analysis window removes punishment-related fluctuations from analysis. (**C**) Pupil size is similar across V-rule bimodal and A-rule bimodal segments (pupillometry recorded in *n*=14 sessions, 5 mice), suggesting similar levels of arousal and task engagement across rules. Difference box plots: central line: median; box edges: 25th and 75th percentiles; whiskers: data points not considered outliers. (**D**) Min-max-normalized locomotion is also similar across rules.

The online version of this article includes the following figure supplement(s) for figure 2:

**Figure supplement 1.** Pupil size and locomotion compared between unimodal and bimodal blocks.

## Single unit recording in AC

After mice learned the AV rule-switching task, a craniotomy was made over right AC, to allow for acute recordings during behavior using multichannel probes spanning the full cortical depth (*Figure 3A*). In total, we recorded AC activity in 10 mice during 23 behavioral sessions meeting inclusion criteria. The putative cortical depth of each sorted single unit (SU) was assigned by calculating the fractional position of the channel with the largest waveform amplitude within the span of channels in AC, as estimated from spontaneous and tone-evoked recordings following the task (*Figure 3B*). A separate

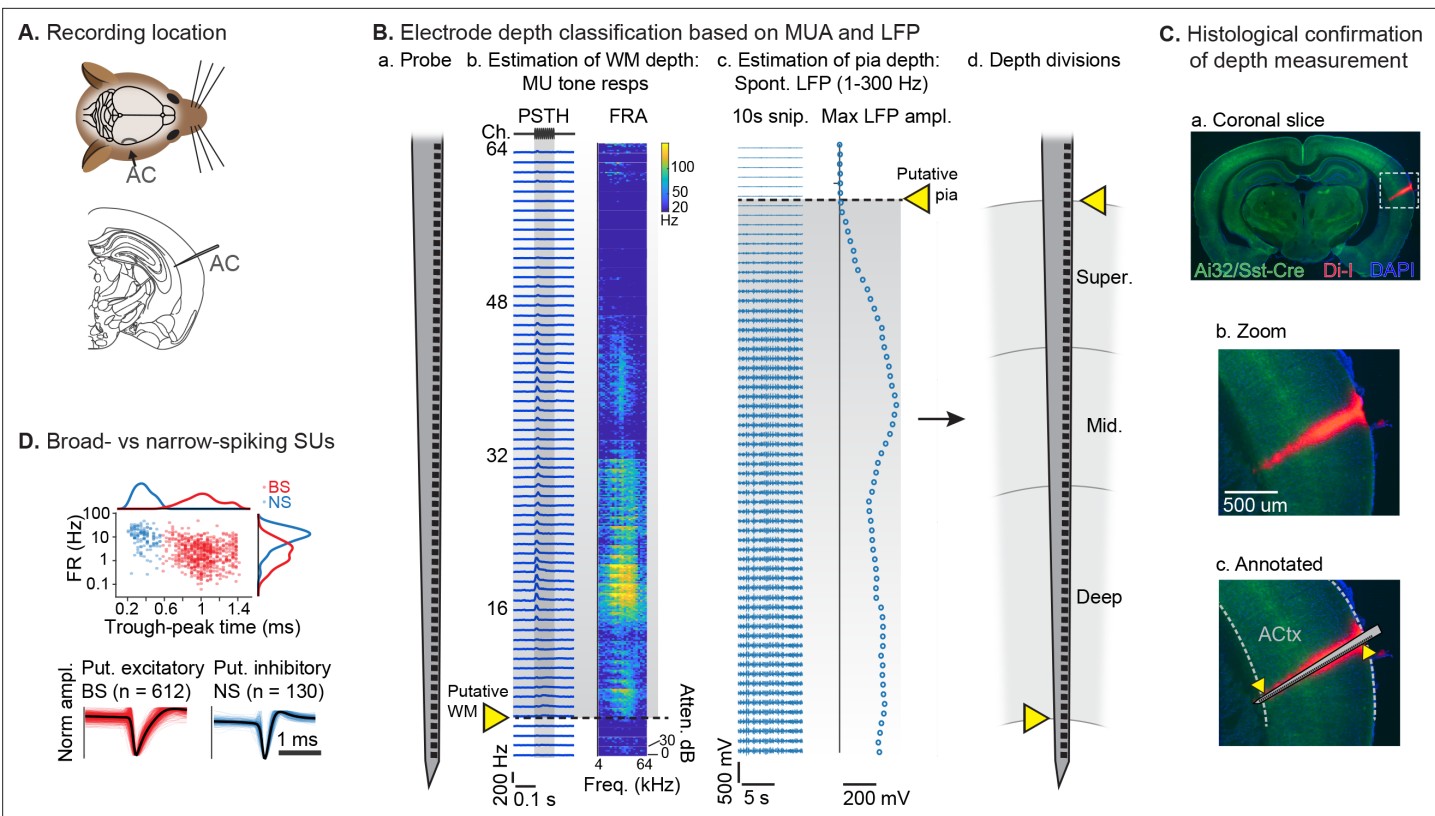

**Figure 3.** Single unit (SU) recording and depth estimation in auditory cortex. (**A**) Translaminar probes were used to record activity in right auditory cortex (AC). (**B**) Physiological estimation of cortical depth. (**a**) Linear 64-channel probe captures all activity in layers of AC. (**b**) Example tone-evoked multi-unit (MU) sound responses by channel, providing a marker for the border of deep cortex and white matter (WM). Left: peristimulus time histogram (PSTH) plots showing mean tone response by time. Right: frequency response area (FRA) shows mean response during tone stimulus by frequency/attenuation. MU responses poorly estimate the upper cortical boundary due to low somatic spiking activity in the superficial cortex. (**c**) Local field potential (LFP; 1–300 Hz filtered) provides a marker for the upper cortex-pia boundary. Left: 10 s snippet of LFP by channel. Right: maximum LFP amplitude by channel, with putative pia location defined as the first deviation from probe-wise minimum LFP amplitude. (**d**) Channels are assigned cortical depths based on fractional division of cortex into 'superficial', 'middle', and 'deep', with fractions based on supragranular, granular, and infragranular anatomical divisions. (**C**) Histological confirmation of cortical depth estimation technique. (**a**) Coronal slice showing DI-I probe track (red) in right AC. Green: eYFP fluorescence from Ai32/Sst-Cre mouse strain. Blue: DAPI stain to visualize cell bodies. (**b**) Zoomed area indicated by dashed rectangle in a. (**c**) Probe overlay and WM/pia boundaries. Yellow arrows indicate locations of physiologically determined cortical span from B, showing close correspondence with Di-I probe track. (**D**) Sorted SU waveforms were divided into narrow-spiking (putative fast-spiking inhibitory) and broad-spiking (putative excitatory) based on a waveform trough-peak time boundary of 0.6 ms.

set of experiments to visualize probe tracks with the fluorescent dye Di-I provided support for this depth estimation technique (*Figure 3C*; *DiCarlo et al., 1996*; *Morrill and Hasenstaub, 2018*). We then divided the fractional depth values into superficial, middle, and deep groups, approximating the supragranular, granular, and infragranular laminae. We further divided SUs into narrow-spiking (NS, putative inhibitory; *n*=130, 18%) and broad-spiking (BS, predominantly excitatory; *n*=612, 82%) populations based on trough-peak time (*Figure 3D*; *Bigelow et al., 2019*; *Cardin et al., 2007*; *Nandy et al., 2017*; *Phillips et al., 2017*).

## Modality-selective attention modulates stimulus-evoked firing rates

To measure the effects of modality-selective attention on stimulus processing in AC, we began by comparing SU responses to bimodal decision stimuli across task rules. These responses reflected physically identical stimuli and similar levels of arousal and locomotion, as shown in *Figure 2*. Activity patterns evoked by decision stimuli and modulatory effects of task rule were diverse (*Figure 4A*). To capture a predominantly sensory-driven response component, we measured mean firing rates (FRs) during the first 300 ms post-stimulus onset (*Figure 4B*), which preceded most lick responses (lick latency median: 611 ms; 5–95th percentiles: 289–1078 ms; 5.4% of licks <300 ms, *n*=2852

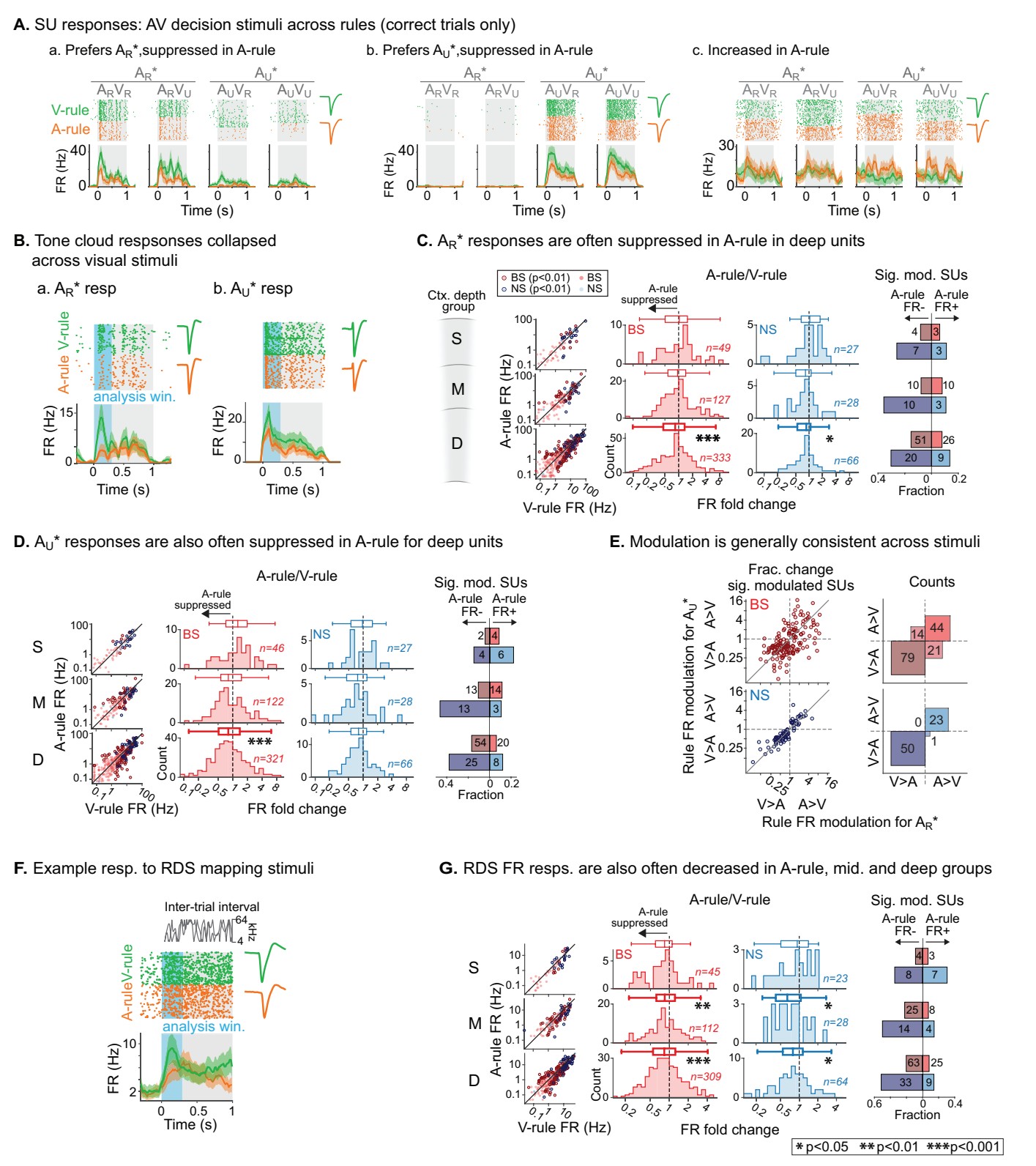

**Figure 4.** Net suppression of sound-evoked firing rates during auditory attention. (**A**) Example single unit (SU) responses to physically identical audiovisual (AV) stimuli across task rules ($A_R^*$=$A_RV_R$ and $A_RV_U$ collapsed; $A_U^*$=$A_UV_U$ and $A_UV_R$ collapsed). (**a**) Response showing preference for $A_R^*$ tone cloud, suppressed in A-rule relative to V-rule. (**b**) Preference for $A_U^*$ tone cloud, suppressed in A-rule. (**c**) Moderately enhanced firing rate (FR) in A-rule. (**B**) Example SU responses to (**a**) $A_R^*$ and (**b**) $A_U^*$ tone clouds (TCs). Early sensory-driven response analysis window (0–0.3 s) shown in light blue. (**C**) Group

*Figure 4 continued on next page*

*Figure 4 continued*

data: responses to TCs rewarded in A-rule ($A_R*$) between rules by unit type and depth. Scatter plots (left) show FR across rules. Red: broad-spiking (BS) units. Blue: narrow-spiking (NS) units. Outlined: significantly modulated units, paired t-test, Benjamini-Hochberg false discovery rate (FDR)-adjusted, q=0.01. Fold change histograms show A-rule FR divided by V-rule FR for all units; bins to the left of 1 (dashed line) indicate FR suppression in A-rule. Box plots above histograms: central line: median; box edges: 25th and 75th percentiles; whiskers: data points not considered outliers. Asterisks indicate FDR-adjusted (q=0.05, n=6 tests) p-values from paired Wilcoxon signed-rank tests of mean FRs across rules; no asterisk: not significant (p>0.05). Right: fractions of significantly modulated units (inclusion as described above) over total. Darker colors indicate fractions with significantly suppressed FRs in A-rule; lighter colors indicate enhanced FRs in A-rule. (**D**) Responses to TCs unrewarded in A-rule ($A_U*$). All conventions as in C. (**E**) Comparison of unit FR modulation by rule between $A_R*$ (abscissa) and $A_U*$ (ordinate). Top: BS units, bottom: NS units. Scatter plots (left) show all units with significant rule modulation for $A_R*$, $A_U*$, or both. Modulation values <1 indicate suppressed FR response in A-rule. Note the increased density of units below 1 for BS and NS units. Right: counts of units by direction of FR rule modulation. Most units lie in quadrants with similar direction of modulation across stimuli, suggesting that attentional effects on FR are not frequency- or stimulus identity-dependent. (**F**) Example SU response to the onset of the random double sweep (RDS) mapping stimulus, showing analysis window for calculating FR (0–0.3 s, blue). (**G**) Group data for RDS FR modulation across rules by depth and BS/NS classifications. All conventions as in C.

The online version of this article includes the following source data for figure 4:

**Source data 1.** Decision stimulus response firing rate (FR) across rules.

**Source data 2.** Random double sweep (RDS) response firing rates (FR) across rules.

total lick trials across dataset). Trials with licks earlier than 300 ms were excluded from analysis. We first compared A-rule and V-rule responses to the TC rewarded in the A-rule ($A_R*$: $A_R V_R$ and $A_R V_U$ responses combined). Averaging across units, responses in the deep layers were suppressed in the A-rule relative to the V-rule for both NS and BS units (*Figure 4C*; deep BS: p=2.8e-4, Z=4.1, median fold change [FC; A-rule/V-rule]: 0.89, n=333 SUs; deep NS: p=0.011, Z=2.9, median FC: 0.87, n=66; paired WSR, FDR-adjusted p-values; see *Figure 4—source data 1A* for full stats). No significant group-level change was found in middle or superficial units. Consistent with group-level trends, individual units with significant FR decreases in the A-rule (p<0.01, unpaired t-test) substantially outnumbered units with significant FR increases for all unit populations other than superficial and mid-depth BS units (*Figure 4C*, right).

A similar pattern of attention-related modulation was observed for unrewarded stimuli in the A-rule ($A_U*$: $A_U V_R$ and $A_U V_U$ responses combined). At the group level, superficial and middle unit responses did not differ significantly between conditions, whereas deep BS units were suppressed in the A-rule (*Figure 4D*; deep BS: p=2.0e-06, Z=5.11, median FC: 0.81, n=321; paired WSR, FDR-adjusted p-values; see *Figure 4—source data 1B* for full stats). Relative fractions of units with significantly modulated FRs to $A_U*$ stimuli were similar to those described above for $A_R*$ stimuli (*Figure 4D*, right), with the exception of the superficial group, in which slightly more units had increased FRs. We further found that most units showed the same direction of modulation for $A_R*$ and $A_U*$ stimuli (*Figure 4E*), with a similar modulation sign observed for 78% of BS units (50% suppressed for both $A_R*$ and $A_U*$, 28% enhanced for both) and 99% of NS units (68% suppressed for both, 31% enhanced for both). These findings suggest that modality-selective attention similarly influences FRs evoked by task-relevant target and distractor sounds with different acoustic properties and learned behavioral values.

To determine whether these attentional influences might generalize to task-irrelevant sounds, we examined responses to RDS sounds presented during the ITI. Using the same analysis window (300 ms post-stimulus onset, *Figure 4F*), we found that attention-related modulation of FR responses evoked by task-irrelevant sounds was highly similar to that observed for both types of decision stimuli: middle- and deep-layer BS and NS populations exhibited group-level FR suppression during the A-rule (*Figure 4G*), whereas superficial layer units showed no difference (middle BS: p=6.0e-3, Z=3.1, median FC: 0.85, n=112; middle NS: p=0.014, Z=2.7, median FC: 0.65, n=28; deep BS: p=1.2e-6, Z=5.2, median FC: 0.84, n=309; deep NS: p=0.021, Z=2.5, median FC: 0.80, n=64; paired WSR; see *Figure 4—source data 2* for full stats). Significantly modulated unit counts were again highly biased toward suppression in the A-rule, with pronounced differences in the middle and deep unit groups (*Figure 4G*, right). Together, these results show that auditory-selective attention tends to reduce FR responses to sounds, regardless of their behavioral relevance, valence, or spectral content, and that these influences are strongest for deep-layer units.

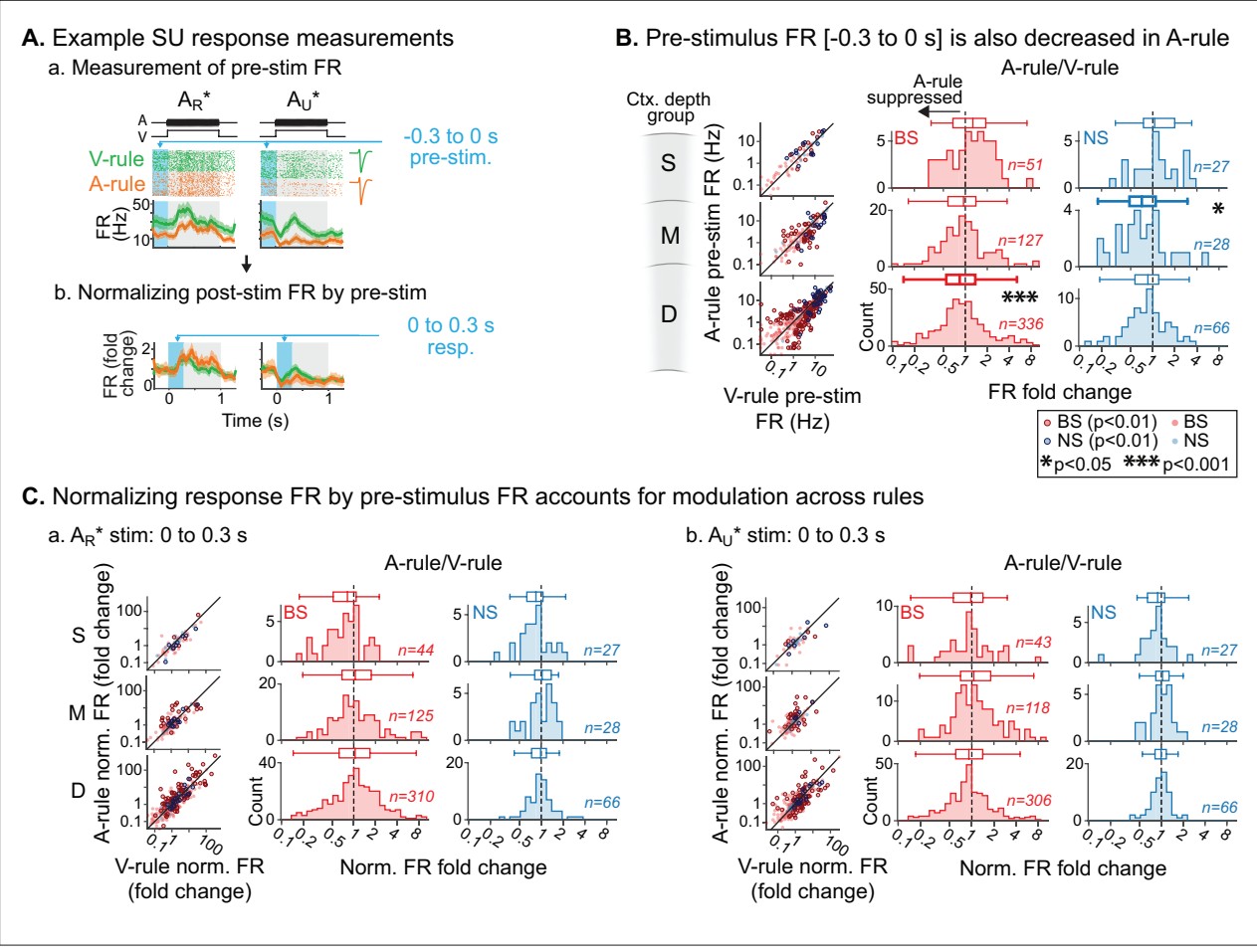

**Figure 5.** Attention-related modulation of sound-evoked responses largely reflects pre-stimulus activity changes. (**A**) Example pre-stimulus firing rate (FR) measurement, and normalization of post-stimulus response. (**a**) Raw FR by condition and stimulus. Pre-stimulus analysis window shown in blue (−0.3–0 s). (**b**) Normalized FRs (FR divided by mean pre-stimulus FR). (**B**) Group data: pre-stimulus onset FR compared across rules, with data organized by depth (S=superficial, M=middle, D=deep) and broad-spiking/narrow-spiking (BS/NS) (red/blue). Conventions as in *Figure 4*. Scatter plots (left) show individual units, with significantly modulated units outlined (paired t-test, Benjamini-Hochberg false discovery rate (FDR)-adjusted, q=0.01). Difference histograms show A-rule/V-rule for all units shown in scatters; fold change <1 indicates suppression during the A-rule. As in *Figure 4*, asterisks represent p-values from FDR-adjusted paired Wilcoxon signed-rank tests on each group (q=0.05, n=6 tests). Absence of asterisk: not significant. (**C**) Group data: response as fold change, normalized by pre-stimulus FR. Conventions as in B and *Figure 4*. After accounting for pre-stimulus modulation, effects of rule on FR are abolished.

The online version of this article includes the following source data for figure 5:

**Source data 1.** Pre-stimulus firing rate (FR) across rules.

**Source data 2.** Decision stimulus baseline-adjusted firing rate (FR) across rules.

## Modality-selective attention also modulates pre-stimulus FRs

Previous studies have found that modulation of ongoing activity in sensory cortex can influence subsequent sensory-evoked responses (*Arieli et al., 1996*; *Haider and McCormick, 2009*). Thus, the response suppression during auditory attention reported above may either reflect specific decreases in stimulus responsivity or general decreases in ongoing activity. To address these possibilities, we quantified FRs in a pre-stimulus window spanning 300 ms prior to decision stimulus onset in which no sounds were presented (*Figure 5A*.a). Although this window may include anticipatory modulation of activity (*Cox et al., 2019*; *Egner et al., 2010*; *Samuelsen et al., 2012*), it nevertheless provides a measure of baseline activity for comparison with evoked responses. We observed significant group-level decreases in pre-stimulus FRs during the A-rule for units in the middle NS and deep BS groups, but no modulation of superficial units (*Figure 5B*; middle NS: p=0.039, *Z*=2.48, median FC: 0.71, n=28; deep BS: p=4.2e-05, *Z*=4.49, median FC: 0.87, n=336; paired WSR, FDR-adjusted p-values; see

*Figure 5—source data 1* for full stats). To test whether the reduction in pre-stimulus FR was sufficient to account for stimulus-evoked changes reported above, we recalculated FRs evoked by decision stimuli as FC from pre-stimulus FRs (*Figure 5A*.b). Following this adjustment and after FDR correction, the middle- and deep-layer unit population responses no longer differed between rules (*Figure 5C*; *Figure 5—source data 2*). Together, these results suggest that group-level decreases in evoked FRs during A-rule are largely due to generalized suppression of ongoing AC activity.

## Attention-related suppression is driven by units without STRF tuning

We next sought to determine whether attention-related changes in stimulus response were related to the tuning preferences of units, a phenomenon termed 'feature attention' previously observed in both monkey VC (*Maunsell and Treue, 2006*; *Treue and Martínez Trujillo, 1999*) and AC (*Da Costa et al., 2013*). The RDS mapping stimulus, which we have previously used to efficiently identify auditory response properties and AV interactions (*Bigelow et al., 2022*), was used to generate spectrotemporal receptive fields (STRFs) through reverse correlation (*Figure 6A*; *Aertsen and Johannesma, 1981*; *de Boer, 1968*; *Gourévitch et al., 2015*). Tuning for each STRF was measured through a trial-to-trial reliability metric, which we used to divide units into those with activity changes that were reliably evoked by defined spectral or temporal features (tuned STRFs, $n=172$; *Figure 6C*.a) and those without feature-evoked changes (untuned, $n=409$; *Figure 6C*.b). Spiking activity levels were higher in tuned units compared to untuned (*Figure 6D*.a). To control for possible activity level-dependent effects, we compared our population of tuned units to a randomly selected subset of untuned units which was matched for both sample size and FR to the tuned population (*Figure 6D*.b). We then examined attentional modulation of stimulus responses between the tuned and subsampled untuned groups. Responses to the rewarded TC ($A_R^*$), unrewarded TC ($A_U^*$), and the RDS mapping stimuli were significantly modulated by task rule in the untuned group, but not the tuned group (tuned: all $p≥0.18$, all $|Z|≤1.72$; untuned: all $p≤0.023$, all $|Z|≥2.27$; one-way WSR vs. modulation of 1 [equal across rules], FDR-adjusted p-values; *Figure 6—source data 1C, D*). Nevertheless, comparisons across these tuned and untuned groups showed that the distributions did not significantly differ after multiple comparisons correction (all $p≥0.12$, all $|Z|≤2.06$; WSR).

An important caveat is that the RDS stimuli may not capture all units with some degree of tuning preference. As such, a conservative interpretation would be that group-level suppression during auditory attention is driven by units that do not exhibit strong tuning preferences. Additionally, both tuned and untuned populations contained units with significant evoked responses to the two TCs, although fractions of responsive units were higher in the tuned group (*Figure 6D*.c). This shows that an absence of STRF tuning does not imply that units were not responsive to the task stimuli.

For the tuned group, does frequency preference determine degree of attentional modulation? We measured the best frequency (BF) of the excitatory field in each tuned STRF (*Figure 6F*). Consistent with previous work showing that task demands shape frequency representation in AC (*Atiani et al., 2009*; *Fritz et al., 2005*; *Fritz et al., 2003*; *Yin et al., 2014*), we found a strong BF preference for a 1-octave band around the center frequency of the rewarded TC (*Figure 6G*). Furthermore, distributions of BFs measured during the A-rule and V-rule were strikingly similar. This suggests that in our task, AC had shifted its frequency representation in a manner that was not rule-dependent. To test whether modulation by rule was dependent on tuning, we next divided units by their BF, as measured from the A-rule STRF, into groups near center frequency of $A_R$ (±0.5 octaves), near $A_U$ or with a BF outside of either band. No difference between the tuning groups was observed for responses to $A_R^*$ or $A_U^*$ TCs (Kruskal-Wallis non-parametric ANOVA, all $p>0.12$, all $H<5.5$, FDR-adjusted p-values; *Figure 6—source data 1C, D*), suggesting that frequency tuning does not determine suppression or enhancement by attention in this task.

## Attention to sound increases encoding efficiency in deep-layer BS units

Previous work has established that FR changes do not necessarily imply changes in the amount of information spikes carry about sensory stimuli. For instance, optogenetic activation of inhibitory interneurons can reduce FRs in AC without changing information, suggesting increased encoding efficiency (*Phillips and Hasenstaub, 2016*). By contrast, locomotion reduces both FRs and information in AC (*Bigelow et al., 2019*). To determine whether reduced FRs evoked by decision stimuli were accompanied by changes in information or encoding efficiency, we used a peristimulus time

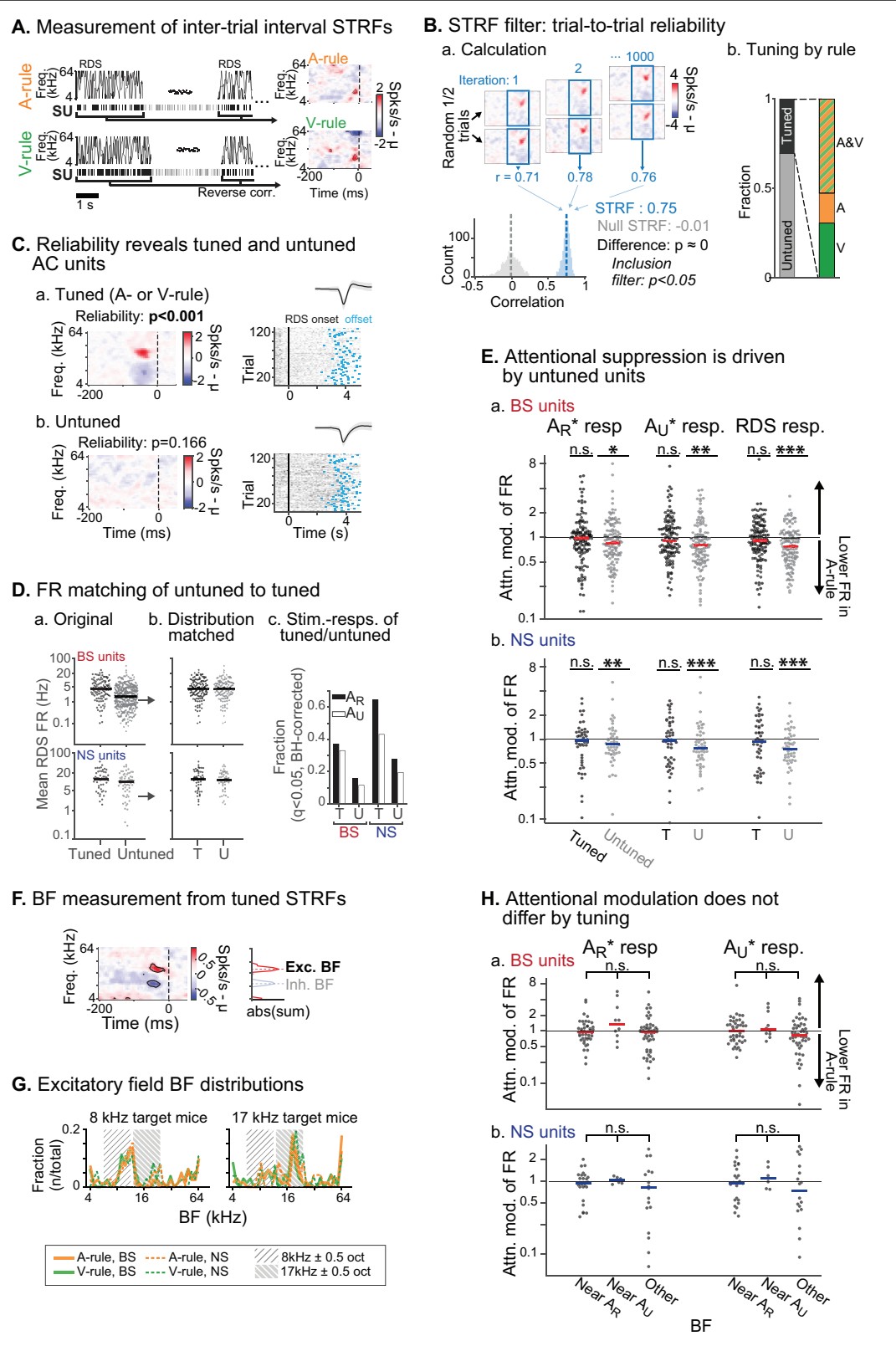

**Figure 6.** Attentional modulation of spike rate is driven by neurons without spectrotemporal receptive field (STRF) tuning. (**A**) STRFs for A-rule and V-rule were calculated from spikes during the inter-trial-interval random-double sweep (RDS) stimulus using standard reverse correlation methods. (**B**) STRF reliability as a measure for tuning. (**a**) Reliability was measured through correlations of randomly subsampled halves of all RDS presentations,

*Figure 6 continued*

repeated 1000 times. A p-value was calculated empirically through comparison of correlation value distributions from the actual STRF and a null STRF, generated from random circular shuffling of spike trains relative to stimulus. (**b**) Left: fraction of tuned and untuned units. Right: fractions of units with STRF tuning in both A- and V-rules (AV), A-rule only (A), or V-rule only (V). (**C**) Trial-to-trial reliability metric separates AC units into those with tuned STRFs (**a**) and untuned STRFs (**b**). (**D**) To control for activity level-driven effects, the larger group with untuned STRFs (*n*=345, 64 for broad-spiking [BS], narrow-spiking [NS]) is matched for sample size and firing rate to the group with tuned STRFs (*n*=121, 51 for BS, NS). (**a**) Mean firing rate (FR) during RDS mapping stimulus by tuned and untuned groups. (**b**) FR distribution-matched groups. (**c**) Tuned group contains a larger share of decision stimulus-responsive units compared with untuned (for both $A_R$* and $A_U$* tone clouds [TCs]). Stimulus responsiveness is defined as a significant FR difference between 0.3–0 s pre-stimulus window and 0–0.3 s post-stimulus window, paired t-test, Benjamini-Hochberg false discovery rate (FDR)-adjusted, q=0.01. (**E**) Untuned unit group is suppressed during auditory attention, while tuned unit group is not. (**a**) Attentional modulation of BS unit responses for task decision stimuli (left: $A_R$*; right: $A_U$*) and RDS mapping stimuli (right). Paired Wilcoxon signed-rank between mean FR in A-rule and V-rule, FDR-corrected at q=0.05 (*n*=3 comparisons per group). Asterisks indicated FDR-adjusted p-values. (**b**) NS, conventions as in **a**. Asterisks indicate significance: *p<0.05; **p<0.01; ***p<0.001. (**F**) Measurement of best frequency (BF) from tuned STRF group, based on peaks of absolute values of significant time-frequency bins summed across time (–100 ms to 0 window). Significant time-frequency bins (p<0.01) determined by comparison of observed STRF values with distribution of values from spike time-shuffled null STRF. (**G**) BFs of excitatory STRF fields show that AC units are preferentially tuned near the center frequency of the target (rewarded) TC. (**H**) Attentional modulation by BF of tuned units: tuned near $A_R$ (BF ± 0.5 octaves from TC center), $A_U$, or tuned to frequency outside either band. Response modulation does not differ by BF tuning for any comparison ($A_R$* or $A_U$* response and BS or NS units; Kruskal-Wallis test; BS: all p>0.11, NS: all p>0.81, FDR-adjusted).

The online version of this article includes the following source data for figure 6:

**Source data 1.** Stimulus response modulation across rules by spectrotemporal receptive field (STRF) tuning.

---

histogram (PSTH)-based neural pattern decoder to compare sound discrimination across attentional states (*Foffani and Moxon, 2004*; *Hoglen et al., 2018*; *Malone et al., 2007*). For each unit, the decoder generates a single-trial test PSTH and then compares these to two or more template PSTHs from different stimulus response conditions, generated sans test trial (*Figure 7A*). The test trial is assigned to the template that is closest in *n*-dimensional Euclidean space, reflecting *n* PSTH bins. This is repeated for all trials, generating new templates for each classifier run. After all trials have been classified, a confusion matrix is generated. From this, we calculated accuracy of classification, MI (bits), and encoding efficiency, a spike-rate-normalized MI (bits/spike). As in previous analyses, a 0–300 ms post-stimulus onset window was used in this method to restrict decoding to a predominantly sensory-driven component of the response. The binwidth for generating PSTHs was 30 ms (*Hoglen et al., 2018*). Only trials with correct responses (hits and CRs) and units with a minimum stimulus response FR of 1 Hz to both stimuli used in the decoder comparison were included.

We found that task rule could be decoded at greater than chance levels from responses to all four AV stimuli, and at all depth and NS/BS groups, showing that attentional state modulates decision stimulus PSTH responses throughout AC (*Figure 7—figure supplement 1*; *Figure 7—source data 1*). These comparisons suggest response modulation by task rule, but do not address how information processing changes *across* the rules. To test this, we used the decoder to compare accuracy in discriminating between responses to $A_R$* (rewarded in A-rule) and responses to $A_U$* (unrewarded in A-rule) bimodal stimuli across A-rule and V-rule conditions. This mimics the TC discrimination required by the mice during the A-rule. In both rules, classification accuracy for the auditory decision stimuli ($A_R$*, $A_U$*) was higher than chance for all depth and BS/NS groups (see scatter plots in *Figure 7B*; all p≤1.4e-05, all |Z|≥4.2, one-way WSR vs. chance [50%]; see *Figure 7—source data 2A* for stats). Sound classification accuracy ($A_R$*, $A_U$*) did not significantly differ *across* the A-rule and V-rule (*Figure 7B*, $A_R$* vs. $A_U$* *comparison across rules:* all p≥0.17, all |Z|≤1.39, see *Figure 7—source data 3A* for full stats; paired WSR on decoder accuracy in A-rule vs. V-rule by depth and NS/BS groups). Despite a reduction in activity levels during auditory attention, there was no loss in decoder accuracy, suggesting a possible change in encoding efficiency.

Through analysis of all decoder runs, we found that classifier accuracy and raw information were indeed correlated with FR (accuracy: $r(3001)=0.49$, p=2.3e-180; MI: $r(3001)=0.41$, p=1.5e-123;

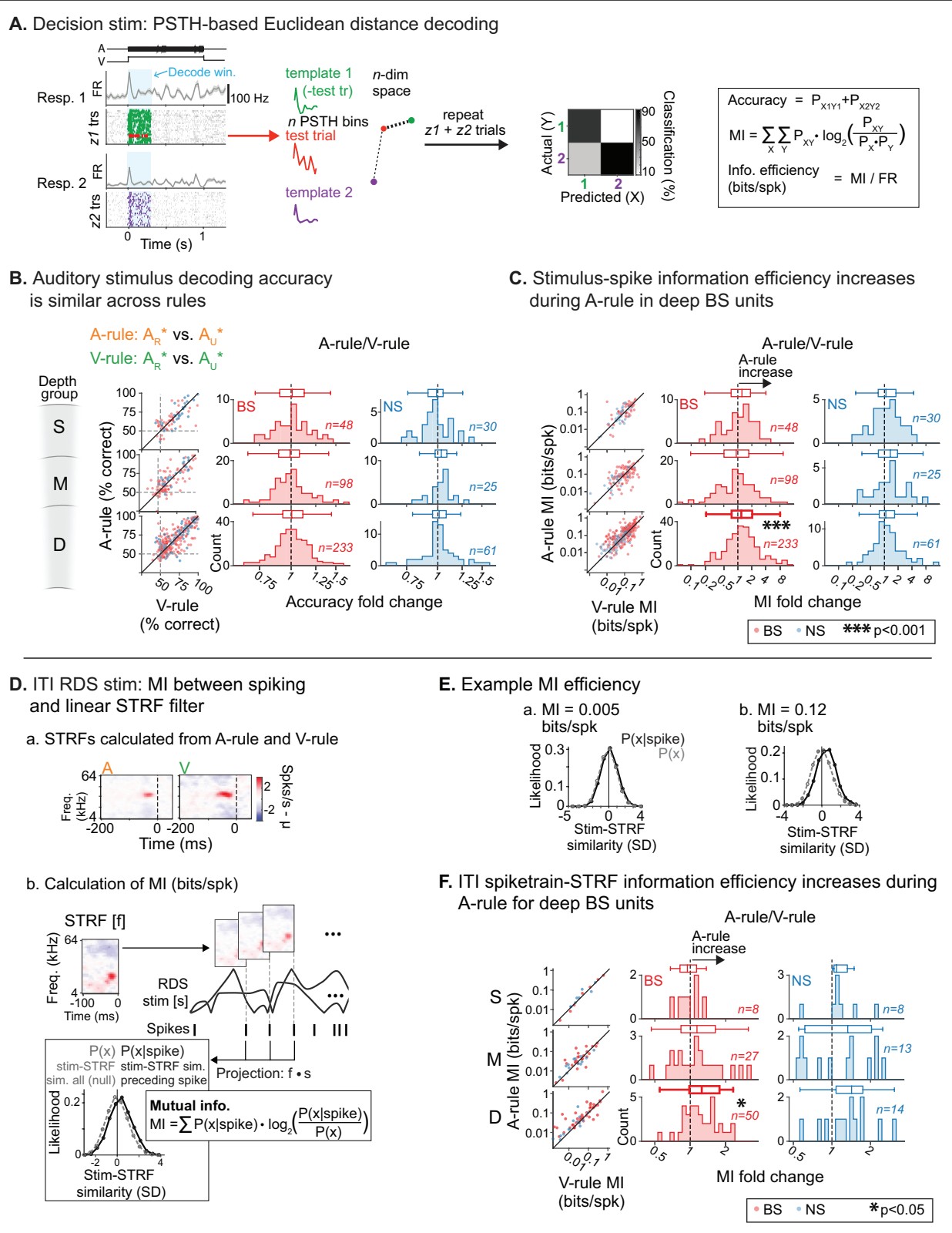

**Figure 7.** Auditory attention increases sound encoding efficiency in deep-layer broad-spiking units. (**A**) Peristimulus time histogram (PSTH)-based spike train decoding analysis. Time-binned responses for each single trial (test trial; red) are compared to PSTHs (templates; green, purple) reflecting responses to each stimulus averaged across all other trials. Trials are classified as belonging to the template nearest to the test trial in *n*-dimensional Euclidean space (*n*=number of PSTH bins). A confusion matrix (right) reflecting predicted/actual outcomes for all trials is used to calculate accuracy,

*Figure 7 continued on next page*

*Figure 7 continued*

mutual information (MI; bits), and encoding efficiency (bits/spike). (**B**) Decoding accuracy of auditory stimulus identity, compared across attentional states. Decoder setup mimics task faced by mice in the A-rule: discrimination between $A_R^*$ and $A_U^*$ tone cloud identities. Results represent average of two decoder runs, which differ in their paired visual stimulus, but yield similar results: $A_RV_R$ vs. $A_UV_R$ and $A_RV_U$ vs. $A_UV_U$ (see *Figure 7—figure supplement 2* for separate presentation of these data). Conventions as in *Figure 4C*. and elsewhere. Scatter plots represent decoder accuracy from individual units; dashed lines show chance level (50%). Histograms show raw unit counts for A-rule/V-rule fold change. S=superficial; M=middle; D=deep. No change in accuracy is observed across rules. (**C**) Stimulus-spike information efficiency (bits/spike, calculation shown in A) for PSTH-based decoding increases for deep broad-spiking units during auditory attention. Conventions as in **B**. (**D**) Measuring encoding changes across attentional states for inter-trial interval (ITI) mapping stimuli. (**a**) Example spectrotemporal receptive fields (STRFs) calculated from ITIs of A-rule and V-rule from the same single unit (SU). (**b**) Estimation of mutual information efficiency of ITI random double sweep (RDS) stimuli: the STRF is convolved with the windows of the RDS stimulus to define two distributions of relative STRF-stimulus similarity values: 1. *P(x|spike)*, from time windows preceding a spike, and 2. *P(x)*, a null distribution from non-overlapping time windows tiling the full stimulus duration. Information encoding efficiency is calculated as shown, reflecting the divergence between these distributions, which increases when spiking preferentially occurs during periods of higher stimulus-STRF similarity. Mutual information (MI) values are calculated from STRFs in A-rule and V-rule separately. (**E**) Example of spike train-STRF encoding efficiency from two SUs: low (**a**) and high (**b**) bits/spike examples. (**F**) Comparison of spike train-STRF encoding efficiency across rules, showing increased encoding efficiency in A-rule for deep broad-spiking units.

The online version of this article includes the following source data and figure supplement(s) for figure 7:

**Source data 1.** Decoding of task rule from stimulus response peristimulus time histogram (PSTHs).

**Source data 2.** Decoder accuracy by rule.

**Source data 3.** Comparison of decoding accuracy across rules.

**Source data 4.** Mutual information (MI) efficiency across rules.

**Source data 5.** Comparison of spectrotemporal receptive field (STRF) mutual information (MI) (bits/spike) across rules.

**Figure supplement 1.** Decoding of task rule from stimulus response peristimulus time histograms (PSTHs).

**Figure supplement 2.** Decoding and information efficiency changes across rules are similar across visual stimulus pairings.

**Figure supplement 3.** Deep auditory cortex (AC) A-rule information efficiency increases are driven by firing rate (FR) suppressed units.

Pearson's correlation, all $A_R^*$ vs. $A_U^*$ decoder runs). Thus, normalizing information by mean joint per-trial spike rate for the two responses in each decode (bits/spike) provides insight into the efficiency with which spikes are used to represent stimuli. We found that this encoding efficiency measure increased by ~20% during the A-rule for deep-layer BS units (*Figure 7C*, $A_R^*$ vs. $A_U^*$ *comparison across rules:* deep BS: p=2.9e-04, Z=−4.06, paired WSR, FDR-adjusted p-value; median FC: 1.19 [fold change: A-rule/V-rule]; V-rule: 0.15±0.13, A-rule: 0.19±0.19, mean bits/spike ± SDs, n=233; all other groups p≥0.40, all |Z|≤1.47; see *Figure 7—source data 4A* for full stats). No other unit subpopulations showed significant changes. Note that for clarity, the above results are presented as the mean of decoder comparisons $A_RV_R$ vs. $A_UV_R$ and $A_RV_U$ vs. $A_UV_U$, thus collapsing visual stimulus identity. Analysis of these comparisons separately yields highly similar results (*Figure 7—figure supplement 2*; *Figure 7—source data 2–4*), suggesting that visual stimulus identity does not contribute substantially to decoder accuracy or encoding efficiency at the level of group analysis.

## Receptive fields mapped during the ITI also show increased stimulus encoding efficiency

The analyses above revealed that auditory attention increased the per-spike encoding efficiency of task decision sounds. Does this effect of cross-modal attention switching generalize to encoding of sounds that were explicitly designed to be task-irrelevant? This helps determine whether attention observed here is specific to features of the auditory stream or broadly alters encoding of incoming auditory information. To address this, we tested whether information between STRFs derived from task-irrelevant ITI sounds and spike trains was modulated by attentional demands of the task. We restricted our analyses to only those units with STRFs passing the reliability criterion shown in *Figure 6B*. To calculate STRF-spike train MI for each SU, we first calculated probability distributions of STRF-stimulus projection values for all stimulus time points (*P(x)*) and for those time points preceding a spike (*P(x|spike)*; *Figure 7D*). Intuitively, these projection values reflect the similarity between a windowed stimulus segment at a given timepoint and the STRF. The divergence of the two projection distributions is captured in a spike-rate-normalized MI measure (bits/spike; encoding efficiency), which describes the reliability with which spikes are determined by stimulus features of the STRF (*Figure 7E*).

No differences in encoding efficiency between conditions were observed in the superficial or middle BS/NS groups, or the deep NS group. Instead, consistent with our earlier findings for decision stimuli, encoding efficiency showed a significant A-rule increase in the deep BS subpopulation (*Figure 7F*; deep BS: p=0.014, $Z$=–3.05, median FC: 1.25, $n$=50; paired WSR, FDR-adjusted p-value; FC: A-rule/V-rule; mean bits/spike ± SDs; all other groups p≥0.24, all $|Z|$≤1.66; see *Figure 7—source data 5* for full stats). This finding shows that during auditory attention, stimulus encoding is better described by a linear STRF filter and thus better tracks physical sound features. Furthermore, it suggests that increased encoding efficiency resulting from decreased spiking is a general effect of auditory attention in deep-layer BS units, regardless of the context-based behavioral relevance or learned valence of the sounds.

## Information encoding efficiency changes are driven by suppressed units

The increase in A-rule encoding efficiency and decrease in average FRs in deep AC led us to further explore the relationship between activity level and information changes. Specifically, we tested whether group-level information efficiency changes are driven by SUs with suppressed responses, and how the minority of units with increased A-rule FRs perform in the decoder. We therefore examined classifier accuracy and encoding efficiency for target and distractor ($A_R$* vs. $A_U$*) decoding separately for deep-layer BS units with increased and decreased FRs in the A-rule (*Figure 7—figure supplement 3*). We found that units with increased FRs (39%; $n$=96) exhibited a significant increase in A-rule decoding accuracy (*Figure 7—figure supplement 3C*; p=0.0030, $Z$=–2.97, med. FC: 1.04, V-rule % correct: 66.4±15.5, A-rule: 69.5±15.5, $n$=96; paired WSR; FC: A-rule/V-rule; mean ± SDs), but no significant change in encoding efficiency (p=0.84, $Z$=0.2, V-rule bits/spike: 0.18±0.15, A-rule: 0.18±0.16). By contrast, units with suppressed FRs (60%; $n$=146) showed no significant change in decoding accuracy (*Figure 7—figure supplement 3D*; p=0.44, $Z$=0.77, V-rule: 67.32±15.25, A-rule: 66.53±14.61), but a 44% increase in encoding efficiency (p=1.8e-07, $Z$=–5.22, median FC: 1.44, V-rule: 0.13±0.12, A-rule: 0.18±0.19; paired WSR). These results suggest that the minority of units that increase FRs in the A-rule perform marginally better at decoding the auditory stimulus, and that the units that decrease FRs drive the shift in encoding efficiency.

## Attention-related FR changes predict correct task performance

To ensure that mice were adequately engaged and attentive in the task, the analyses described above excluded any trials in which the incorrect behavioral response was made. However, an examination of these error trials, which may correlate with lapses in attention, could provide insight into the moment-to-moment behavioral relevance of the attentional effects described above. We have shown that attention to sound is marked by a net suppression of pre-stimulus and evoked FRs. We hypothesized that, if this attentional modulation is behaviorally meaningful, FRs preceding A-rule error trials may be more similar to sound-unattended V-rule trials than to A-rule correct trials. We addressed this possibility by comparing pre-stimulus FRs in error vs. correct trials (300 ms prior to stimulus onset; *Figure 8B*). Because misses were uncommon (*Figure 8A*), we restricted our analysis to the comparison of FA and CR trials to allow for adequate sampling of each trial outcome. We included only behavior sessions with at least 10 FA and CR trials (A-rule and V-rule trials considered separately). This decreased unit sample sizes ($n$=234, 58 across all depth groups for BS, NS; min. group size = 9, 2 for BS, NS). Given the small sample of NS units and the likelihood of insufficient power, NS units were not included in this analysis. When considering BS units *with increased FRs in the A-rule*, we found no significant group-level difference between A-rule FA and CR trials at any cortical depth (*Figure 8C*; *Figure 8—source data 1A* for full stats; all p≥0.29, all $|Z|$≤1.43, paired WSR, FDR-adjusted p-values). However, deep cortical BS units *with A-rule suppression* showed significantly higher pre-stimulus FRs prior to A-rule FA trials than CR trials (deep BS [$n$=98]: mean FR difference between pre-stimulus FA and CR trials = 0.35 Hz, p=0.0098, $Z$=–3.15; paired WSR; other depth groups: p≥0.28, $|Z|$≤1.48; all p-values FDR-adjusted). This is unlikely to reflect a motor effect of higher FR before a lick, as it was specific to the A-rule: pre-stimulus FRs in A-rule-suppressed or A-rule-enhanced units did not differ between FA and CR trials in the V-rule (*Figure 8C*.c; *Figure 8—source data 1B*; paired WSR: all p≥0.68, all $|Z|$≤1.59, FDR-adjusted p-values). Together, these findings suggest that FR reductions typical of modality-selective attention directly relate to behavioral outcomes.

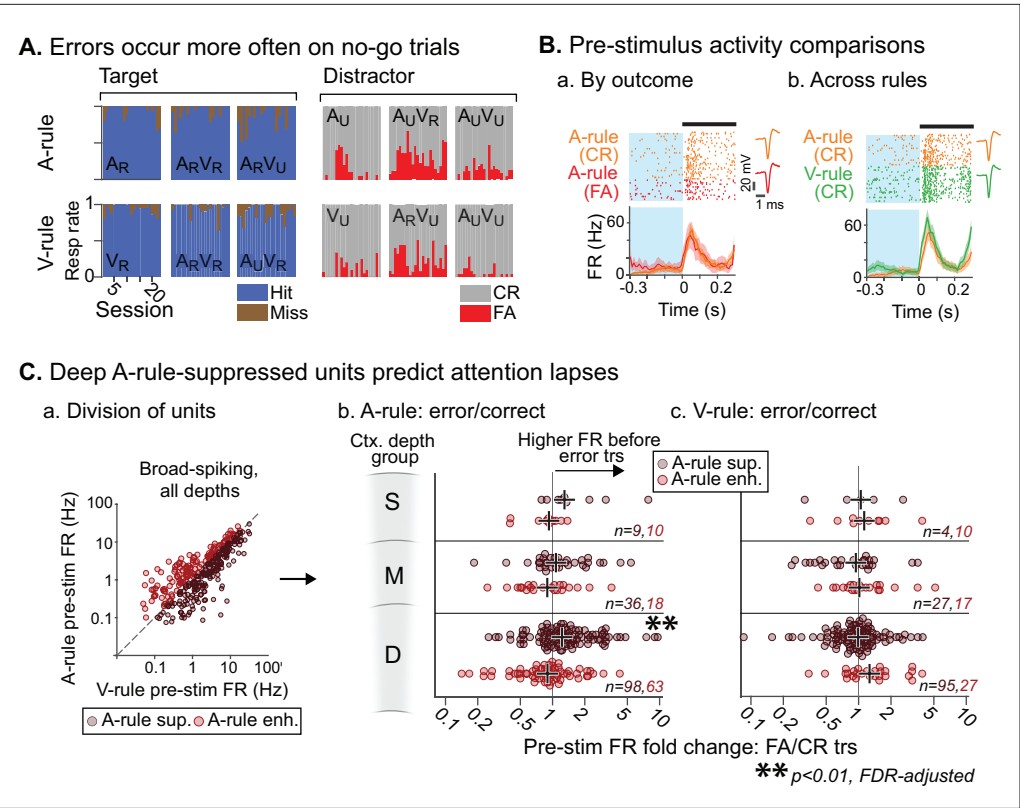

**Figure 8.** Attentionally suppressed units predict behavior performance during auditory attention. (**A**) Summary of behavioral outcomes by session (*n*=23, 10 mice), organized by task stimulus. Bar sequence follows chronology of experiments. Error trials are predominantly false alarms (FAs). To allow sufficient trials for measurement of activity levels across behavioral outcomes, subsequent analysis focuses on analysis of FAs vs. correct rejects (CRs). (**B**) Example unit showing behavioral outcome- and rule-dependent firing rate (FR) modulation. Pre-stimulus FR analysis window (−0.3–0 s) shown in blue. (**a**) Pre-stimulus activity for A-rule FA trials (red) is elevated relative to CR trials (orange). (**b**) In the same unit, pre-stimulus activity is elevated in V-rule CR trials (green) relative to A-rule CR trials (orange). (**C**) Division of units into A-rule-suppressed and A-rule-enhanced groups reveals suppression of activity as a neural signature of correct task performance. (**a**) Broad-spiking (BS) units from sessions with ≥10 FA and CR trials are divided into A-rule suppressed and A-rule enhanced groups. (**b**) Deep units that are suppressed during auditory attention relative to visual show higher firing rates on A-rule error trials relative to correct trials (p=0.0098, paired Wilcoxon signed-rank test, false discovery rate [FDR]-adjusted for *n*=6 tests). Median of group indicated by black cross. No such trend exists for the A-rule-enhanced population. (**c**) Pre-stimulus activity does not predict V-rule behavioral outcomes in the same groups, suggesting that AC activity suppression is related to performance on sound but not visual stimulus discrimination (all p>0.68, paired Wilcoxon signed-rank test, FDR-adjusted for *n*=6 tests).

The online version of this article includes the following source data for figure 8:

**Source data 1.** Pre-stimulus firing rates (FRs) for false alarm vs correct reject trials.

# Discussion

In the present study, we recorded SU activity across AC layers in mice performing an AV rule-switching task. We compared responses evoked by identical stimuli under conditions of auditory or visual modality-selective attention. Attention to sound shifted AC stimulus representation by decreasing activity of untuned units and increasing encoding efficiency in the deep cortical laminae. Pre-stimulus activity was also reduced by auditory attention, which accounted for changes in stimulus-evoked responses. The effects of attention extended beyond the decision stimuli required to complete the task; responses to task-irrelevant receptive field mapping stimuli exhibited similar reductions in evoked activity and increases in encoding efficiency, suggesting that attention to sound induces a stimulus-general shift in processing. This attentional shift was behaviorally meaningful, with error trials in the A-rule predicted by higher FRs in the set of units that is suppressed under auditory attention.

Taken together, these results show that attending to sound results in a general suppression of ongoing activity in AC, while retaining activity critical for sensory representation.

Attentional highlighting of behaviorally relevant signals may employ multiple mechanisms, including response enhancement or noise suppression. Feature selective attention studies have shown that FRs for neurons tuned to attended features are often increased, thereby increasing the reliability of the sensory cortical readout (*Desimone and Duncan, 1995*; *Moran and Desimone, 1985*; *Reynolds and Chelazzi, 2004*). Another mechanism which may act in tandem with response enhancement is the reduction of noise to improve encoding reliability. Noise reduction may act through decreased rates in pre-stimulus baseline activity (*Buran et al., 2014*), reduced variance in single neuron rates (*Mitchell et al., 2007*), or decreased correlations of noise across the population (*Cohen and Maunsell, 2009*; *Downer et al., 2015*). In the present study, we found no evidence for increased signal-to-noise ratio in the FR signal, as shown by the roughly equal stimulus response magnitudes across rules when normalizing for pre-stimulus rate. However, the timing of activity in AC is known to carry substantial information (*Hoglen et al., 2018*; *Krishna and Semple, 2000*; *Malone et al., 2007*), which would not be captured by coarse rate estimations. By accounting for fine scale temporal patterns with a PSTH-based pattern classifier and analysis of stimulus-STRF selectivity, we show that decreased ongoing activity and a concomitant increase in encoding efficiency at the group level provides an additional mechanism for attentional noise reduction, perhaps refining the stimulus-encoding portion of the neural signal for readout in downstream brain areas.

Previous studies of behavioral state-dependent state changes in auditory processing have typically compared task-engaged and passive sound processing. While this paradigm does not specifically isolate the effects of attention due to confounds of arousal, attention, reward expectation, and motor activity (*Saderi et al., 2021*), it has provided valuable insight into the dependence of sensory processing on task-engaged behavioral states. Consistent with our findings, this work has shown that AC stimulus-evoked spiking responses are predominantly suppressed during self-initiated task engagement when compared to passive listening (*Bagur et al., 2018*; *Carcea et al., 2017*; *Kuchibhotla et al., 2017*; *Otazu et al., 2009*). Activity levels preceding a stimulus may also decrease (*Buran et al., 2014*; *Carcea et al., 2017*), although some studies in AC do not show this effect (*Otazu et al., 2009*). Reductions of pre-stimulus activity during task engagement have also been observed in rat gustatory cortex (*Yoshida and Katz, 2011*) and monkey visual cortex (*Bisley and Goldberg, 2003*; *Cox et al., 2019*; *Herrington and Assad, 2010*; *Sato and Schall, 2001*).

Neuronal stimulus preferences relative to a target have been shown to determine the degree of attentional modulation, such that stimulus-evoked responses for attended features are generally enhanced but can also be suppressed for features outside of the receptive field (*Reynolds and Chelazzi, 2004*). Here, we find that frequency preferences of units with STRF tuning do not appear to determine suppression or enhancement within the task, but critically we also find that the bulk of units with STRF tuning exhibit a preference for frequencies near the rewarded TC (*Figure 6G*). This is consistent with a body of work from Shamma, Fritz, and colleagues showing that engagement in an auditory discrimination task rapidly shifts AC receptive fields to enhance frequency representation of behaviorally relevant stimuli (*Atiani et al., 2009*; *Fritz et al., 2005*; *Fritz et al., 2003*; *Yin et al., 2014*). In our task, mice were trained for multiple months prior to physiological recordings, and the TC frequencies of rewarded and unrewarded stimuli were held consistent for each animal. As such, spectral representation in the AC of our highly trained mice is biased toward task-relevant stimuli. Speculatively, it is possible that tuning-dependent attentional modulation may occur in earlier stages of task acquisition, but that the substantial reconfiguration of sound processing tailored to the task alters its expression after training. The distribution of preferred frequencies also does not shift between auditory and visual rules, suggesting that attending to visual stimuli does not place plasticity-inducing demands on AC frequency representation. Instead, we find that units without STRF tuning drive the reduction in neural activity during auditory attention. An important caveat is that our STRF-based approach is only one way to determine AC tuning, and other stimulus and analysis methods may reveal additional tuning preferences. Nevertheless, we believe that this method provides a useful classification for degree of tuning. This result is also consistent with our information theoretic analyses in that both suggest that attention to sound may selectively remove spikes that are minimally sound-driven.

As in previous studies, attention-related modulation was not uniformly expressed across cortical depths and neuron types. Changes in both FR and encoding efficiency were most prominent in deep-layer neurons. These findings extend several previous studies reporting larger effects of attention in infragranular LFP and multi-unit activity (MUA) (*O'Connell et al., 2014*; *Zempeltzi et al., 2020*). These physiological outcomes are consistent with anatomical work suggesting that top-down modulatory signals arrive primarily in the supragranular and infragranular layers (*Felleman and Van Essen, 1991*). As the main cortical output layer, information shifts in the infragranular population would differentially influence subcortical sites and other cortical regions (*Salin and Bullier, 1995*). One important caveat is that superficial AC is known to have lower spontaneous and evoked FRs than deeper cortex (e.g., *Figure 4C*; *Christianson et al., 2011*; *Sakata and Harris, 2009*), which may have made it more difficult for us to observe statistically significant attention-related effects. Furthermore, although we tried to minimize neural tissue damage through technical considerations such as using a slow probe insertion speed (*Fiáth et al., 2019*), the superficial layers likely sustain the greatest level of damage when the probe is inserted to span the full cortical depth. Despite these factors, we were able to isolate a reasonably large sample size of responsive neurons in superficial cortex from successful behavior sessions (*n*=119 units, of which 57% were stimulus-responsive). Nevertheless, we cannot rule out whether the absence of observed attentional modulation at superficial depths may have been due to experimental limitations such as the comparatively small sample size. Future work employing imaging techniques to target superficial neurons may help resolve this.

Previous studies have reported larger effects of task engagement or attention in inhibitory interneurons (*Kuchibhotla et al., 2017*; *Mitchell et al., 2007*). As such, attention-related reduction of activity could be sustained by inhibitory network drive. Our approach of dividing activity into BS and NS did not suggest a general increase in NS activity during auditory attention. However, we observed heterogenous types of modulation; in many units, NS activity decreased during auditory attention, but in a smaller group, there was a significant increase. An important caveat is that the BS/NS distinction is an imperfect approximation of excitatory/inhibitory activity, with many inhibitory cell types presenting a BS waveform phenotype (e.g., somatostatin-positive interneurons; *Li et al., 2015*). An alternative mechanism is that excitatory drive is decreased during auditory attention. These two proposed mechanisms – increased inhibitory tone and decreased excitatory drive – are not mutually exclusive.

Our findings suggest that attentional selection is achieved by removal of a noise background on which sound stimulus-encoding activity sits. This is in line with an influential theory of cortical attention that posits that spontaneous activity fluctuations partly reflect internal processes such as mental imagery or memory recall, in contrast with activity that arises from external sensory stimulation (*Harris and Thiele, 2011*). In this model, attention suppresses internally generated spontaneous activity to favor the processing of behaviorally relevant external stimulation. The work presented here offers multiple pieces of evidence in favor of this theory. Auditory attention suppresses activity in untuned units, affecting both pre-stimulus and stimulus-evoked activity. This activity reduction does not alter stimulus-spike train decoding accuracy, but instead increases stimulus encoding efficiency and preserves stimulus representation.

In summary, we demonstrate a novel connection between attention-induced shifts in activity levels and stimulus encoding in early sensory cortex, which are directly related to behavioral outcomes. Previous research suggests that such effects reflect top-down control by executive networks comprising frontal, parietal, thalamic, and striatal areas (*Cools et al., 2004*; *Crone et al., 2006*; *Licata et al., 2017*; *Rikhye et al., 2018*; *Rougier et al., 2005*; *Toth and Assad, 2002*; *Wimmer et al., 2015*). These networks may act as a context-dependent switch, routing attentional modulatory feedback to the sensory systems. In the present study, we provide evidence that such modulation specifically suppresses stimulus-irrelevant spiking, thus enhancing encoding efficiency in deep AC neurons.

# Materials and methods
## Animals
All experiments were approved by the Institutional Animal Care and Use Committee at the University of California, San Francisco. Twenty-seven C57BL/6 background male mice were surgically implanted with a headpost and began behavioral training, of which 10 completed the training and successfully performed the task during physiology recording sessions. All mice began the experiment between

ages P56 and P84. Mice used in this report expressed optogenetic effectors in various subsets of interneurons, which we intended to use for optogenetic identification of cells (*Lima et al., 2009*; analysis not included here). These mice were generated by crossing an interneuron subpopulation-specific Cre driver line (PV-Cre JAX Stock Nr. 012358; Sst-Cre: JAX Stock Nr. 013044) with either the Ai32 strain (JAX Stock Nr. 012569), expressing Cre-dependent eYFP-tagged channelrhodopsin-2, or the Ai40 strain (JAX Stock Nr. 021188), expressing Cre-dependent eGFP-tagged archaerhodopsin-3. Of the 10 behavior mice included in this report, 6 were Ai32/Sst-Cre, 3 were Ai32/PV-Cre, and 1 was Ai40/Sst-Cre. In most experiments (*n*=21 recordings), brief, low-level optogenetic pulses during the ITI of the task were used to identify opsin-expressing neurons (<0.3 mW light; 5 light pulses of 10 ms duration, every ~1.5 min); these analyses are outside of the scope of this report. The optogenetic stimulation protocol was consistent through A- and V-rules of the task. Unit stimulus response FRs and behavioral response error rates were not statistically different between trials immediately after optogenetic pulses and stimulus-matched trials preceding the pulses.

All mice were housed in groups of 2–5 for the duration of the behavioral training until the craniotomy. Post-craniotomy and during physiology recordings, mice were housed singly (up to 6 days) to protect the surgical site. Mice were kept in a 12 hr/12 hr reversed dark/light cycle. All training occurred during the dark period, when mice show increased activity and behavioral task performance (*Roedel et al., 2006*).

## AV rule-switching behavior task

Adult mice (>P56) were trained on an AV go/no-go rule-switching behavior task. In this task, mice were positioned on a floating spherical treadmill in front of a monitor and a speaker, and an optical computer mouse recorded treadmill movement. Mice licked to receive a reward depending on auditory, visual, or AV stimulus presentation ('decision' stimuli, either 'target' or 'distractor'), but the modality predictive of the reward changed partway through the behavioral session. Each session would start with a unimodal go/no-go block, in which a series of auditory ($A_R$, $A_U$; 17 or 8 kHz TC) or visual ($V_R$, $V_U$; upward or rightward moving gratings) stimuli was presented. After stimulus presentation, mice signaled choice by either licking a spout in front of the mouth or withholding licking. Licking at the target unimodal stimulus would trigger a water reward, while licking at the distractor would trigger a short dark timeout. After a fixed number of unimodal trials, the stimuli would become AV, but the rule for which stimulus modality predicted reward would carry over from the unimodal block. All four stimulus combinations ($A_RV_R$, $A_RV_U$, $A_UV_R$, $A_UV_U$) would be presented in the AV block, such that two AV combinations would be target stimuli and two would be distractor. Then, after completing a fixed number of trials in the AV block, the task using the rule of the opposite modality would begin; a unimodal block with the other modality would start, followed by a second AV block using the rule from the preceding unimodal block. For any mouse, the stimuli predictive of the reward in each rule was kept constant across days and training sessions (e.g., a 17 kHz TC would always predict a reward in the A-rule, and a rightward grating would always predict a reward in the V-rule).

The task was self-paced using a virtual foraging approach, in which mouse locomotion (measured through treadmill rotation) would cause a track of randomly placed dots on the screen to move downward. After a randomly varied virtual distance, a decision stimulus would be presented, at which point the mouse would lick or withhold licking to signal choice. For receptive field mapping during physiology experiments, an RDS stimulus was presented in-between decision stimuli, during the inter-trial track portion. Stimuli are detailed below.

## Behavior training and apparatus

Prior to any training, mice were surgically implanted with a stainless steel headplate, used both for head fixation during the task and for physiology recordings after the task was learned (surgical methods described below). Three days post-implant, mice began a water restriction protocol based on previously published guidelines (*Guo et al., 2014*). Throughout the course of training, mice received a minimum water amount of 25 mL/kg/day, based on weight at time of surgical implant. After recovery from surgery, mice were given ~7 days to adjust to water restriction. Then, mice were head fixed and habituated to the floating treadmill for 15–30 min daily sessions with no stimulus presentation for 2–3 days. After mice appeared comfortable on the treadmill, a phased behavioral task training regimen began. Mice were trained once daily for ~6 days per week. On day 1, mice were introduced

to an auditory-only (A-only) stimulus training version of the task in which $A_R$ ('target'/'rewarded') or $A_U$ ('distractor'/'unrewarded') stimuli were presented, and a reward would be automatically administered shortly after the onset of $A_R$. Next, the mice were put on an operant version of the A-only task, which required licking any time after the onset of $A_R$ to receive a reward and withholding of licking during $A_U$ to avoid a dark timeout punishment. Mice achieved proficiency, defined as 2 or more consecutive days of sensitivity index $d'>1.5$ (see *Data analysis* for calculation), on the A-only task after 11.0±4.7 days after start of training (median ± SD, $n=10$ successful mice). Then, a similar training structure was repeated for the visual task: V-only stimulus training with automatic rewards for $V_R$, but not $V_U$, followed by an operant version of the visual task requiring licks for rewards (median time to proficiency: 26.0±7.2 days after start). After learning the tasks for each modality separately, mice were introduced to an auditory-AV (A-AV) version, in which the rule from the auditory stimulus carried over to the AV block. This was intermixed with training days on a visual-AV (V-AV) version of the task. Number of training days on A-AV or V-AV were decided based on prior performance, with extra training given as needed. Mice were considered proficient at this stage after performing with $d'>1.5$ on each rule (A-AV; V-AV) on 2 consecutive days (median time to proficiency: 40.0±15.8 days after start). Finally, the full rule-switching task was introduced (*Figure 1D*), generally alternating between days of V-rule-first and the A-rule-first task sequences but allocating more training days to task orders as needed. Because physiology recordings were acute and strictly limited to 6 days after craniotomy, we set a greater threshold for expert-level performance on the full task before advancing to physiology: 3 consecutive days of $d'>2.5$ (median time to expertise: 90.5±31.8 days). Care was taken to train each mouse at a roughly consistent time of day (no more than ~1–2 hr day-to-day variation). During expert-level task performance, mice typically completed 260–300 trials in a daily session (30 A-only; 100–120 A-AV; 30 V-only; 100–120 V-AV).

The behavior training setup was controlled by two computers: a behavior monitoring and reward control PC (OptiPlex 7040 MT, Dell) and a dedicated stimulus presentation machine running Mac OS X (Mac Mini, Apple). Stimulus presentation was controlled with MATLAB using custom software (https://github.com/HasenstaubLab/AVtrainer-stim/tree/main/demo; copy archived at swh:1:rev:737720f41fd5302b90fd5e60a10822270381818c;path=/demo/; *Morrill et al., 2022*), and inter-machine communication used the ZeroMQ protocol. Auditory and visual stimuli were generated and presented using the Psychophysics Toolbox Version 3 (*Kleiner et al., 2007*). Water rewards were administered using a programmable syringe pump (NE-500, New Era Pump Systems, Farmingdale, NY), positioned outside of the sound-attenuating recording chamber. Early in training, water reward volume was set at 0.01 mL per correct response, but over training the reward volume was gradually decreased to 0.006 mL to achieve greater trial counts. Licking events were recorded using a custom photobeam-based lickometer circuit based on plans provided by Evan Remington (Xiaoqin Wang Lab, Johns Hopkins University). Licks were registered when an IR photobeam positioned in front of the lick tube was broken, queried at a sample rate of 100 Hz by an Arduino Uno microcontroller (Arduino, LLC).

## In vivo awake recordings during behavior

Animals in this experiment underwent two surgeries: first, before training a surgery to implant a custom steel headplate over the temporal skull using dental cement was conducted. The animal was anesthetized using isoflurane and a headplate was implanted over AC, ~2.5 mm posterior to bregma and under the squamosal ridge, to allow for physiology recordings after achieving task expertise. When mice completed the training regimen outlined above, a craniotomy surgery was performed. The animal was again anesthetized using isoflurane and an elliptical opening (0.75 mm wide × 1.5 mm long) was made in the skull over AC using a dental drill. This opening was promptly covered with silicone elastomer (Kwik-Cast, World Precision Instruments), and the animal was allowed to recover overnight. The following day, the animal was affixed by its headplate over the treadmill inside of a sound-attenuating recording chamber, the silicone plug over the craniotomy was removed, and the craniotomy was flushed with saline. A silver chloride ground wire was placed into the craniotomy well at a safe distance from the exposed brain. A 64-channel linear probe (20 µm site spacing; Cambridge Neurotech, Cambridge, UK) was slowly inserted in the brain using a motorized microdrive (FHC, Bowdoin, ME) at an approximate rate of ~1 µm/s (*Fiáth et al., 2019*). After reaching the desired depth, the brain was allowed to settle for 10 min, after which the water spout, lickometer,

visual stimulus delivery monitor, and speaker were positioned in front of the mouse, and the behavior session commenced. Behavior sessions were sometimes stopped early and restarted due to poor performance. In approximately half of behavior-physiology sessions (13 of 23 successful recordings), the task was stopped due to low performance after the rule transition and restarted at the beginning (unimodal block) of the second rule. To control for possible effects of task order, attempts were made to counterbalance recordings from A-rule first (15) and V-rule first (8) behavior sessions.

After completion of the behavior task, the water spout and lickometer were removed, and a series of auditory and/or visual passive experiments were conducted in order to characterize the response properties of the recording site. All stimuli were presented with the auditory and visual stimulation apparatus described above. Following completion of these experiments, the probe was slowly removed, and the brain was covered with a thin layer of freshly mixed 2.5% agarose in saline, followed by a layer of silicone elastomer. The animal was returned to its home cage, and the following day the physiological recording process was repeated. Recordings were made for up to 6 days after the craniotomy. The neural signal acquisition system consisted of an Intan RHD2000 recording board and an RHD2164 amplifier (Intan Technologies), sampling at 30 kHz.

## Auditory and visual stimuli

In-task auditory decision stimuli were 1 s TCs, consisting of 50 ms tone pips overlapping by 25 ms, with frequencies in a 1-octave band around either 17 or 8 kHz. TCs were frozen for the duration of the task, so that each mouse always heard the same pip sequences, allowing for direct comparisons of sound-evoked neural responses across rules without concern that stimulus peculiarities may be driving observed differences. TCs were presented at 60 dB SPL. Visual decision stimuli consisted of a circular moving grating stimulus (33° diameter subtended visual space), which appeared at the center of the screen for 1 s (coincident with TC stimulus during bimodal presentation). Gratings moved either upward or rightward with a 4 Hz temporal frequency, 0.09 cycles/degree spatial frequency at 50% contrast. In-between decision stimulus presentations, an RDS stimulus was presented for receptive field mapping (*Bigelow et al., 2022*; *Gourévitch et al., 2015*). The RDS comprised two uncorrelated random sweeps that varied continuously and smoothly between 4 and 64 kHz, with a maximum sweep modulation frequency of 20 Hz. RDS stimuli were presented at 50 dB SPL.

After the behavior task, passive auditory search stimuli (pure tones, click trains) were presented to characterize response properties of the electrode channel. Click trains consisted of broadband 5 ms white noise pulses, presented at 20 Hz for 500 ms duration. Pure tone stimuli consisted of 100 ms tones of varied frequencies (4–64 kHz, 0.2 octave spacing) and sound attenuation levels (30–60 dB in 5 dB linear steps), with an interstimulus interval of 500 ms.

Auditory stimuli were presented from a free-field electrostatic speaker (ES1, Tucker-Davis Technologies) driven by an external soundcard (Quad-Capture or Octa-Capture, Roland) sampling at 192 kHz. Sound levels were calibrated using a Brüel & Kjær model 2209 meter and a model 4939 microphone. Visual stimuli were presented on a 19-inch LCD monitor with a 60 Hz refresh rate (Asus VW199), positioned 25 cm in front of the mouse and centered horizontally and vertically on the eyes of the mouse. Monitor luminance was calibrated to 25 cd/m² for a gray screen, measured at approximate eye level for the mouse.

## Data analysis
### Behavioral performance
Task performance was evaluated by calculation of the *d′* sensitivity index:

$$d' = Z(H) - Z(F)$$

where *H* is hit rate and *F* is false alarm rate, and *Z* is the inverse normal transform. Because this transform is undefined for values of 0 or 1 and hit rates of 1 commonly occurred in this study, we employed the log-linear transformation, a standard method for correction of extreme proportions, for all calculations of *d′* (*Hautus, 1995*). In this correction, a value of 0.5 is added to all elements of the 2×2 contingency table that defines performance such that:

$$H = (hits + 0.5)/(hits + misses + 1)$$

$$F = (FA + 0.5)/(FA + CR + 1)$$

where *FA* is the false alarm count and *CR* is the correct reject count. To ensure that mice properly transitioned between task rules, *d'* values were calculated separately for responses in the A-rule and the V-rule. Behavioral sessions during physiological recording with *d'*<1.5 in either rule were excluded from analyses, as were any sessions with an FAR >0.5 to stimuli with conflicting reward valances across rules: $A_U V_R$ in A-rule or $A_R V_U$ in V-rule (*n*=23 successful sessions, *n*=10 mice; 1 session excluded due to recording artifact, see below).

## Spike sorting and unit stability evaluation

Spikes were assigned to unit clusters using KiloSort2 (KS2; *Pachitariu et al., 2016*). Clusters were first evaluated for isolation quality through the automated KS2 unit classification algorithm and then with a custom MATLAB interface. In this second step, clusters with non-neuronal waveforms or 2 ms refractory period violations >0.5% were removed from analysis (*Laboy-Juárez et al., 2019*; *Sukiban et al., 2019*). To evaluate stability, activity for each unit was plotted for the recording duration as a raster and binned spike counts (2 min bins) and manually examined for periods with a substantial dropoff in FR (periods flagged for instability: 88 ± 10% [mean ± SD] decrease in FR from median activity level). Flagged unstable periods were marked and removed from analysis (101/742 SUs with flagged durations >10% of recording time). One session meeting behavior performance criteria was excluded due to a high degree of electrical noise contamination.

## Classification of units by depth and waveform shape

Probes with electrode spans of 1260 μm were used, allowing for channels below and above AC. During recording, the probe was lowered to a point where several channels showed a prominent drop in field potential amplitude and spiking activity, indicating penetration into the white matter (*Land et al., 2013*). After behavior sessions, a set of auditory and visual stimulation protocols was used to map response properties of each electrode site, and MUA responses were analyzed. Here, we define MUA as threshold crossings of 4.5 SD above a moving window threshold applied to each channel. Analysis of MUA was restricted to site characterization and is not included in the main results. We analyzed each tone or click PSTH for reliable responses, which we defined as trial-to-trial similarity of p<0.01 (*Escabí et al., 2014*). We designated the deepest channel with a reliable MUA sound response of any magnitude as the deep cortex-white matter border. Limited somatic spiking in the top layer of cortex prevented the use of MUA as a reliable marker for the superficial cortex-pia border (*Senzai et al., 2019*), so we instead relied on an LFP-based measure. To define the top border of cortex, the maximum spontaneous LFP (1–300 Hz) amplitude of a 10 s snippet from each channel was plotted, and the channel at which LFP amplitude dropped off to the approximate probe-wise noise floor (i.e., minimum LFP amplitude) was considered the top channel in cortex (*Figure 3B*.c). These measures were confirmed histologically through Di-I probe marking experiments with a separate group of untrained mice; histology methods described below and elsewhere (*Morrill and Hasenstaub, 2018*). Marking the top and bottom cortical borders generated a span of channels putatively within AC. This span was used to divide channels into superficial, middle, and deep groups, based on measurements of the fraction of cortex attributed to supragranular (layers 1–3), granular (layer 4), and infragranular (layers 5–6) in the mouse AC (Allen Institute Mouse Brain Atlas; https://mouse.brain-map.org/). SUs were assigned the fractional depth of the channel on which the largest magnitude waveform was recorded.

Clusters were also classified into BS (putatively excitatory) and NS (putatively fast-spiking inhibitory) units on the basis of the bimodal distribution of waveform peak-trough durations (*Figure 3D*; NS/BS transition boundary = 0.6 ms). From sessions with successful behavior, we recorded 742 SUs from all cortical depths, comprising 17.5% (130) NS units and 82.5% (612) BS units.

## FR analysis and trial filters

To compare FR responses to stimuli across task rules and to the receptive field mapping stimulus, we measured FR in the first 300 ms post-stimulus onset. Only units with nonzero FRs in both rules were included. To ensure that measurements were capturing periods of task engagement, all trials with incorrect responses (misses and FAs) were excluded from all decision-stimulus analyses, with the exception of those shown in *Figure 8*. We also excluded trials with recorded licks earlier than the

300 ms post-stimulus onset, or in the 500 ms pre-stimulus onset. Given these filters, analyses were restricted to units present in the recording during at least 10 trials (correct behavioral choice and without 'early licks') for each stimulus type.

## PSTH-based Euclidean distance decoding

A PSTH-based decoder was used to compute the MI between spike trains and stimulus identity (*Figure 7A*; *Foffani and Moxon, 2004*; *Hoglen et al., 2018*; *Malone et al., 2007*). In this method, two or more responses are compared by generating template PSTHs by removing one test trial. This test trial response is also binned into a single-trial PSTH, and then classified as belonging to the nearest template in *n*-dimensional Euclidean space, where *n* is the number of PSTH bins. More formally, the nearest template is that which minimizes the Euclidean norm between test and template vectors (PSTHs). This process is then repeated for all trials comprising the template PSTHs. Decoding accuracy is the percentage of trial responses that are correctly assigned to the stimuli that elicited them. MI is calculated from a confusion matrix of classifications as follows:

$$MI = \sum_i \sum_j P(X_i Y_j) log_2 \left( \frac{P(X_i Y_j)}{P(X_i) * P(Y_j)} \right)$$

where *X* is the decoder prediction, *Y* is the actual, $P(X_i Y_j)$ represents the value of the (*i*, *j*) element of the confusion matrix, and $P(X_i)$ and $P(Y_j)$ are sums on the marginals. This yields a value of MI in bits. To measure encoding efficiency (bits/spike), we normalized MI by the joint mean spikes per trial of the responses submitted to the decoder (*Bigelow et al., 2019*; *Buracas et al., 1998*; *Zador, 1998*).

For consistency with FR analyses, a time window of 0–300 ms, where stimulus onset is 0, was chosen for decoding analysis. A PSTH binwidth of 30 ms was chosen based on optimal binwidth calculations for mouse AC using the same decoding method (*Hoglen et al., 2018*). To filter out units with low responsiveness to any of the stimuli in a given decoding analysis, we required a minimum FR of 1 Hz during the 0–300 ms window in both stimulus conditions. As such, unit sets may differ between each decoding analysis due to units that were responsive to one set of stimuli but unresponsive to others.

## STRF analysis

To test whether task rule modulates auditory receptive fields, we presented an RDS stimulus (described in *Auditory and visual stimuli*) in-between trials for durations of ~1–15 s, depending on rate of task progression. Different randomly generated RDS segments were presented in each ITI, and STRFs were generated separately for each rule. Because total RDS duration varied between the A-rule and the V-rule in a single session, we equated presentation time across rules by truncating the segments of the rule with greater RDS time (presentation time in each rule: 6.8±2.6 min [mean ± SD]; *n*=23 sessions). This ensured that different stimulus presentation times did not bias STRF estimation. The first 200 ms of RDS response was dropped from all STRF analyses to minimize bias from onset transients. SU activity during these short RDS segments was used to generate STRFs for each segment using standard reverse correlation techniques (*Aertsen and Johannesma, 1981*; *de Boer, 1968*; *Gourévitch et al., 2015*). In brief, the spike-triggered average was calculated by summing all stimulus segments that preceded spikes using a window of 200 ms before and 50 ms after each spike. The choice of 200 ms prior to each spike reflects the upper limit of temporal integration times of auditory cortical neurons (*Atencio and Schreiner, 2013*), and the 50 ms post-spike time was included to estimate acausal values, that is, those that would be expected by chance given the stimulus and spike train statistics (*Gourévitch et al., 2015*). STRFs were transformed into units of FR (Hz) using standard methods discussed elsewhere (*Rutkowski et al., 2002*). Units with poorly defined STRFs were filtered out using a trial-to-trial correlation metric (*Escabí et al., 2014*): STRF segments were randomly divided into two halves, re-averaged separately, and a correlation value was calculated for the two STRFs. This process was then repeated 1000 times, and the mean of correlations defined the reliability value for each STRF. We compared the mean observed STRF reliability to a null distribution of reliabilities, generated by repeating the procedure on null STRFs made from circularly shuffled spike trains (preserving spike count and interspike interval but breaking the timing relationship between spikes and stimulus). A p-value was calculated as the fraction of the null STRF reliabilities greater than the mean observed STRF reliability, and STRFs with p<0.05 in either rule were included in subsequent

analyses. Any STRFs from units with greater than 10% of recording duration marked as unstable were removed from analysis.

MI between a spike train and an STRF was measured as the divergence of two distributions: one reflecting the similarity of the windowed stimulus segments (RDS) preceding a spike and the STRF, and the other reflecting the similarity of all possible windowed stimulus segments and the STRF, regardless of whether a spike occurred (*Figure 7D*; *Atencio et al., 2008*; *Atencio and Schreiner, 2012*; *Escabi and Schreiner, 2002*). Stimulus-STRF similarity was defined as the inner product of the STRF and the stimulus segment of equivalent dimensions, with higher values reflecting closer matches between the STRF and stimulus. The distribution $P(z|spike)$ was generated from $z = s \cdot STRF$, where $s$ represents all RDS stimulus segments that preceded a spike. Then the distribution $P(z)$ was made from similarity calculations of all possible windowed RDS segments and the STRF. The mean $\mu$ and the standard deviation (SD) $\sigma$ of $P(z)$ were calculated, and the distributions were transformed into units of SD: $x = (z - \mu)/\sigma$, yielding distributions of $P(x|spike)$ and $P(x)$ expressed in units of SD.

Using the distributions described above, a spike count-normalized measure of MI between the calculated STRF and the spike train can be calculated as:

$$MI = \sum P(x|spike)log_2(\frac{P(x|spike)}{P(x)})$$

We used this value to compare how well STRFs from A-rule and V-rule ITIs predict a spike train, and thus whether activity in each attentional condition is well described by this canonical filter model.

## Statistics

All statistical calculations were performed in MATLAB r2019a and its Statistics and Machine Learning Toolbox, V11.5. For group comparisons of SU responses across task rules, paired WSR tests were used, unless otherwise noted. Because tests were performed separately on each depth and spike waveform subpopulation, the Benjamini-Hochberg FDR procedure was used to correct for multiple comparisons, typically across $n$=6 comparisons (three depth groups, two spike waveform groups; *Benjamini and Hochberg, 1995*). This method relies on controlling the Type I error rate (here, q=0.05), providing increased power over typical family-wise error rate controls. To determine if individual SUs were significantly modulated by rule, an unpaired Student's t-test on FR was used with a threshold of p<0.01. Descriptive statistics reported in text are mean ± standard deviation (SD), unless otherwise noted. Fractional change values between task rules are reported as the median of the A-rule/V-rule. All other statistical tests are described in Results. Sample sizes ($n$) are indicated for each comparison in Results or source data files.

## Histological verification of depth measurement

To test the accuracy of our depth estimation method based on physiological responses (*Figure 3*), we presented the pure tone search stimuli described above to a separate set of untrained control mice while performing extracellular recordings ($n$=11 recordings from four mice; Ai32/Sst-Cre). Before insertion, the probe was painted with the fluorescent lipophilic dye Di-I (*DiCarlo et al., 1996*; *Morrill and Hasenstaub, 2018*). The depth measurement procedure based on physiological signals was carried out as described above, and then probe tracks from each recording were visualized as described previously (*Morrill and Hasenstaub, 2018*). Briefly, after recordings, the animal was euthanized, and the brain was removed and placed into a solution of 4% PFA in PBS (0.1 m, pH 7.4) for 12 hr, followed by 30% sucrose in PBS solution for several days. The brain was then frozen and sliced using a sliding microtome (SM2000R, Leica Biosystems) and slices were imaged with a fluorescence microscope (BZ-X810, Keyence). Di-I probe markings showing cortical depth were consistent with physiological activity-based depth measurements described above (*Figure 3B–C*).

## Acknowledgements

This work was supported by National Institutes of Health grants R01DC014101 and NS116598 to ARH, the National Science Foundation GRFP to RJM, the Klingenstein Foundation to ARH, Hearing Research Inc to ARH, and the Coleman Memorial Fund to ARH.

# Additional information

## Funding

| Funder | Grant reference number | Author |
|---|---|---|
| National Institutes of Health | R01NS116598 | Andrea R Hasenstaub |
| National Institutes of Health | R01DC014101 | Andrea R Hasenstaub |
| National Science Foundation | GFRP | Ryan J Morrill |
| Hearing Research Incorporated | | Andrea R Hasenstaub |
| Klingenstein Foundation | | Andrea R Hasenstaub |
| Coleman Memorial Fund | | Andrea R Hasenstaub |
| National Institutes of Health | F32DC016846 | James Bigelow |

The funders had no role in study design, data collection and interpretation, or the decision to submit the work for publication.

## Author contributions

Ryan J Morrill, Conceptualization, Resources, Data curation, Software, Formal analysis, Validation, Investigation, Visualization, Methodology, Writing – original draft, Project administration, Writing – review and editing; James Bigelow, Conceptualization, Resources, Software, Methodology, Writing – review and editing; Jefferson DeKloe, Investigation, Methodology, Project administration; Andrea R Hasenstaub, Conceptualization, Resources, Formal analysis, Supervision, Funding acquisition, Visualization, Methodology, Project administration, Writing – review and editing

## Author ORCIDs

Ryan J Morrill ⓘ http://orcid.org/0000-0002-8592-4549
Andrea R Hasenstaub ⓘ http://orcid.org/0000-0003-3998-5073

## Ethics

All experiments were approved by the Institutional Animal Care and Use Committee at the University of California, San Francisco.

## Decision letter and Author response

Decision letter https://doi.org/10.7554/eLife.75839.sa1
Author response https://doi.org/10.7554/eLife.75839.sa2

# Additional files

## Supplementary files

• Transparent reporting form

## Data availability

Physiology and behavior data supporting all figures in this manuscript have been submitted to Dryad with https://doi.org/10.7272/Q6BV7DVM.

The following dataset was generated:

| Author(s) | Year | Dataset title | Dataset URL | Database and Identifier |
|---|---|---|---|---|
| Morrill RJ | 2022 | Audiovisual task switching rapidly modulates sound encoding in mouse auditory cortex | https://dx.doi.org/10.7272/Q6BV7DVM | Dryad Digital Repository, 10.7272/Q6BV7DVM |

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
