## [Editor Report]

This is an important paper that is methodologically compelling, harnessing a complex behavioral task for modality-specific control of attention to provide new evidence that directed auditory attention produces a global decrease in auditory cortex firing rates without a loss of stimulus-related information. These findings build on previous results showing that task engagement or locomotion down regulates activity in auditory cortex. The manuscript is comprehensive and well-illustrated. It provides highly detailed analysis of the cortical activity modulations during attentional switching that will be valuable to others within and beyond the field of hearing research.

---

## [Decision Letter]

**Decision letter after peer review:**

Thank you for submitting your article "Audiovisual task switching rapidly modulates sound encoding in mouse auditory cortex" for consideration by *eLife*. Your article has been reviewed by 3 peer reviewers, including Brice Bathellier as Reviewing Editor and Reviewer #1, and the evaluation has been overseen by Andrew King as the Senior Editor.

Essential revisions:

1) The authors should analyze the modulation of auditory responses with respect to frequency tuning of the recorded neurons. Although the target stimuli are tone clouds. The frequency distribution of the cloud is very narrow. The authors should analyze the modulation according to best frequency (e.g. 6 or 8 bins of BFs including the average low and high target frequencies) as measured from the STRF (i.e for neurons that have one). Possibly inhibition is specific to the neurons whose BF is not one of the target, while neurons with BFs near the target tone cloud could be boosted. This is an important piece of information to connect the present study with some publications that show potentiation of the representation of the attended stimulus.

2) Related to point 1. The population of neurons in deep auditory cortex in which firing rates and information content increased were excluded from subsequent analysis with the authors choosing to focus on the 'efficiency' improvement in the suppressed population. It is important to have the analysis in figure 8, which seeks to link changes in firing rate to behaviour, repeated for this population. An alternative explanation (that is potentially supported by the observation in S1 that the firing rate changes are driven mostly by untuned neurons) is that during attention relevant neural populations are enhanced such that their activity is better read out and that homeostatic mechanisms act to suppress other, less informative, activity. It seems likely the authors have the data to rule this in or out.

3) The behavioral data should be presented in a way that better demonstrates how well the mice were selectively attending to the relevant modality. Figure 2 presents the overall performance (d') on this task, which seem robust. However, in figure 8 a more careful analysis of the false alarm rates to the AuVr stimulus (which in the attend-a condition is really the most critical condition to confirm that the animals are performing a selective attention task and not dividing their attention across modalities) look really very high (near 100% in some sessions). Information about the distribution of trial types was apparently not available. If they are not equiprobable (i.e. there are fewer of the conflict trials) then overall d' is a poor measure of selective attention. Rather than an overall d' criterion for performance one that assesses the ability of the animal to selectively attend – such as the false alarm rate on the AuVr attend-a and ArVu on the attend-v – would be more appropriate. In figure 8 it would be helpful to know how the data are ordered – by mouse? The conclusion the reader is meant to take away from figure 8 is that increased firing rate predicts lapses of attention. However enhanced firing could indicate higher arousal and therefore intention to lick. A comparison of correct reject trials and hit trials as well as the hits/fa analysis included would be meaningful here to see whether this is a (pre) motor effect or an attentional effect.

4) The paper relies heavily on Wilcoxon rank sum tests without any multiple comparisons correction. Often the p values are very small and therefore the lack of correction is inconsequential however there are exceptions. Specifically, this is problematic, especially for the pupillometry section where the AVv_v comparison would not survive multiple comparisons (unlike the Ava_a comparison its range includes zero). This begs the question as to whether the significant effect on pupil size simply comes about from the presence of visual stimulus rather than the AV conditions. There is also no justification provided for why this is a one-tailed test.

5) Please also improve captions according to reviewer 2 recommendation. Simplifying the figures according to reviewer 3's recommendations (supplementary figures can be used) may help extend the captions without making them too long.

6) The discussion should be extended according to recommendations of reviewer 2 to include observations from previous studies and of reviewer 3 to include a better discussion of the mechanisms and biological meaning of the present findings (see public review). For the mechanisms, it should be particular attention should be put on the origin of suppression (inhibition or excitation)? The implication of attentional effects before stimulus presentation should also be clarified (referee 3).

7) The authors should evaluate if suppression is more difficult to be picked up in supragranular layers, at the light of the repeated observation (also done by the group of Harris in the primary auditory cortex) that spiking frequencies are significantly lower (often reaching sparse levels, that is close to zero) there.

*Reviewer #1 (Recommendations for the authors):*

1/ The authors should analyze the modulation of auditory responses with respect to frequency tuning of the recorded neurons. Although the target stimuli are tone clouds. The frequency distribution of the cloud is very narrow. The authors should analyze the modulation according to best frequency (e.g. 6 or 8 bins of BFs including the average low and high target frequencies) as measured from the STRF (i.e for neurons that have one). Possibly inhibition is specific to the neurons whose BF is not one of the target, while neurons with BFs near the target tone cloud could be boosted. This is an important piece of information to connect the present study with some publications that show potentiation of the representation of the attended stimulus.

*Reviewer #2 (Recommendations for the authors):*

The Discussion is very short – other studies have found suppressive effects of firing rate alongside maintenance or enhancement of information encoded and that active listening accelerates processing (e.g. Otazu et al., Nat. neuro 2009, Dong et al., J.Neurosci 2013, Town et al., nat. comms. 2018, Kuchibhotla et al., Nat. neuro 2017), but many more show tuning changes. There isn't much attempt to relate the general suppressive effects to changes in tuning in the discussion (e.g. the Shamma lab studies looking at STRF changes due to attention), yet the authors perform relevant analysis in the supplemental materials e.g. S1E, that are not even mentioned in passing anywhere in the current manuscript.

*Reviewer #3 (Recommendations for the authors):*

I would like to highlight that I understand the concept of having a "public" review and an internal one, and I also see that the journal recommends to if possible avoid repetitions, but scientifically I find it not always easy to separate the two parts. With this I mean that I think Authors should address both parts, which also are highly interconnected.

The methodology used is rigorous and the authors document technically in a proper way their main findings. However, one crucial recommendation is to simplify the presentation of the main results by reducing the number of panels per figures to the ones conveying the essential take home message per figure (Figure titles are very clear).

– One important issue is about the layer specificity: how do the authors exclude that suppression could be more difficult to be picked up in supragranular layers, in the light of the repeated observation (also done by the group of Harris in the primary auditory cortex) that spiking frequencies are significantly lower (often reaching sparse levels, that is close to zero) there?

– Another important issue is the role of inhibitory intracortical mechanisms in the observed suppression of spiking. Please clarify in a more explicit way whether the attention-driven mechanisms could be sustained by increased inhibitory drive. In the requested and necessary simplification of data presentation please provide if there is evidence of such inhibitory subnetwork mechanisms. This should be discussed in the light that many interneuron types also do present an "excitatory-regular spiking – like" phenotype. Alternatively, hypothesis concerning possible withdrawal of excitatory drive (e.g. of thalamic origin) should be at least discussed.

– As highlighted above, one main concern is on the drawn conclusion that the reduced spike responses are related to modality-specific attentional mechanisms (as suggested by the fact that they correlate with behavioural accuracy during acoustic but not visual tasks) given that they are also observed during intertrial stimuli upon presentation of task irrelevant stimuli. How is the observation that cross-modal (visually-driven) attentional influences are not reported compatible with the above-suggested view? Please clarify.

---

## [Author Response]

Essential revisions:1) The authors should analyze the modulation of auditory responses with respect to frequency tuning of the recorded neurons. Although the target stimuli are tone clouds. The frequency distribution of the cloud is very narrow. The authors should analyze the modulation according to best frequency (e.g. 6 or 8 bins of BFs including the average low and high target frequencies) as measured from the STRF (i.e for neurons that have one). Possibly inhibition is specific to the neurons whose BF is not one of the target, while neurons with BFs near the target tone cloud could be boosted. This is an important piece of information to connect the present study with some publications that show potentiation of the representation of the attended stimulus.

We have analyzed attentional modulation for neurons with and without significant STRF tuning and found that the suppression of activity during auditory attention is driven by units without tuned STRFs. Among tuned neurons, we did not find any difference in modulation between those with BFs near (± 0.5 octaves) the rewarded tone cloud, near the unrewarded tone cloud, or outside of either range. Importantly, we also found a strong tuning bias for frequencies near the rewarded tone cloud, consistent with the work of Shamma et al., suggesting that during the course of training, there was significant auditory cortical plasticity to better represent features of the task. In the Discussion, we note that “it is possible that tuning-dependent attentional modulation may occur in earlier stages of task acquisition, but that the substantial reconfiguration of sound processing tailored to the task alters its expression after training.” (Discussion lines 453 to 456). This result was included in Figure S1 in the originally submitted version of our manuscript but has now been moved to main Figure 6 and has been included in the Results (lines 230-273) and Discussion (lines 441-464) sections.

2) Related to point 1. The population of neurons in deep auditory cortex in which firing rates and information content increased were excluded from subsequent analysis with the authors choosing to focus on the 'efficiency' improvement in the suppressed population. It is important to have the analysis in figure 8, which seeks to link changes in firing rate to behaviour, repeated for this population. An alternative explanation (that is potentially supported by the observation in S1 that the firing rate changes are driven mostly by untuned neurons) is that during attention relevant neural populations are enhanced such that their activity is better read out and that homeostatic mechanisms act to suppress other, less informative, activity. It seems likely the authors have the data to rule this in or out.

The group of units with increased FRs is now presented in a new version of Figure 8. This also includes new behavioral filters and has multiple-comparisons corrections applied (see our responses to points 3 and 4 below). For the A-rule enhanced group, we observe no difference between pre-stimulus activity on correct versus error trials. In the suppressed group, increased activity is predictive of error trials, suggesting that this is a signature of attentional lapses. This is consistent with our findings that allocation of attention is associated with decreased AC activity. Importantly, in the V-rule, no significant changes between correct and error trials are observed for these groups, indicating that this effect is due to modality-specific attention rather than a global attentional or motor effect.

We agree with the interpretation that during attention, less informative activity is suppressed (see former Figure S1, now Figure 6), although we did not observe selective enhancement of more informative activity. This suggests an attentional mechanism which may broadly employ noise removal more than signal enhancement. The failure of this mechanism would manifest as increased activity during activity, and that is indeed what is reflected in Figure 8.

3) The behavioral data should be presented in a way that better demonstrates how well the mice were selectively attending to the relevant modality. Figure 2 presents the overall performance (d') on this task, which seem robust. However, in figure 8 a more careful analysis of the false alarm rates to the AuVr stimulus (which in the attend-a condition is really the most critical condition to confirm that the animals are performing a selective attention task and not dividing their attention across modalities) look really very high (near 100% in some sessions). Information about the distribution of trial types was apparently not available. If they are not equiprobable (i.e. there are fewer of the conflict trials) then overall d' is a poor measure of selective attention. Rather than an overall d' criterion for performance one that assesses the ability of the animal to selectively attend – such as the false alarm rate on the AuVr attend-a and ArVu on the attend-v – would be more appropriate. In figure 8 it would be helpful to know how the data are ordered – by mouse? The conclusion the reader is meant to take away from figure 8 is that increased firing rate predicts lapses of attention. However enhanced firing could indicate higher arousal and therefore intention to lick. A comparison of correct reject trials and hit trials as well as the hits/fa analysis included would be meaningful here to see whether this is a (pre) motor effect or an attentional effect.

We agree that the behavioral performance on the conflict stimuli (A_U_V_R_ in A-rule and A_R_V_U_ in V-rule) is the most critical for determining modality-selective allocation of attention in this task. As such, we have implemented an additional behavioral filter for inclusion of data. The previous sensitivity index-based filter required *d'*>1.5 for both the A-rule and V-rule, calculated separately. We have also implemented a false alarm rate criteria such that the combined FAR for the two conflict distractor stimuli must be less than 0.5. Given the “go-bias” we observed in this task, in which FAs are more likely errors than misses, this specific metric likely provides a better test of modality-specific attention than *d’* over the entire rule, which includes many stimuli that are consistently at the ceiling for correct performance. Figure 1.F.b reflects these new inclusion criteria using color as an additional dimension on the scatter plot to represent FAR_conflict_. This has moderately reduced the number of included recordings to n = 23 recordings from 10 mice (down from n=27 from 12 mice). This change has not altered any of the main findings from the originally submitted manuscript.

We appreciate the concern that this effect could arise from a (pre) motor source. In Figure 8.C.c. we show that there is no difference between FA and CR pre-stimulus activity in the V-rule, which suggests that this effect is rule-specific and not the result of general motor-driven mechanisms. In addition, we have performed the requested comparison of pre-stimulus activity on hit vs correct reject trials (see Author response image 1). As in Figure 8 of the manuscript, this analysis examines activity over the -300 ms to 0 window relative to stimulus onset. Each dot in the scatters represents a SU, and subplot titles indicate depth grouping as well as p- and Z-values from a paired Willcoxon signed rank (Hit vs CR pre-stim FR). Consistent with the updated manuscript, all p-values are multiple comparisons corrected (see answer below for details). There is no significant difference between Hit and CR pre-stimulus activity in any depth group, suggesting that in our study, correct withholding versus correct licking is not meaningfully predicted by motor preparatory activity alone. This outcome may be expected given random ordering of target (hit) and distractor (CR) trials in our task. Pre-stimulus activity should only be highly predictive of these outcomes if measurable motor preparatory activity in this window outweighed stimulus identity in determining task performance. The high level of performance from trained animals in this task suggests that this is not the case. We have not included this analysis in the manuscript, as Figure 8.C.c may provide a more direct control for the observed effect being driven by motor-related activity.

**Author response image 1. sa2fig1:** 

4) The paper relies heavily on Wilcoxon rank sum tests without any multiple comparisons correction. Often the p values are very small and therefore the lack of correction is inconsequential however there are exceptions. Specifically, this is problematic, especially for the pupillometry section where the AVv_v comparison would not survive multiple comparisons (unlike the Ava_a comparison its range includes zero). This begs the question as to whether the significant effect on pupil size simply comes about from the presence of visual stimulus rather than the AV conditions. There is also no justification provided for why this is a one-tailed test.

We agree that multiple comparisons correction is appropriate for many groups of tests in this study. To achieve this, the resubmission now includes false discovery rate (FDR) control with the Benjamini-Hochberg procedure (see Methods lines 816-821). FDR control has been applied to the outcomes of analyses in which the unit set was divided into depth and waveform-type groups and then tested for the same hypothesis. FDR-adjusted *p­*-values are presented in the text of the paper where noted. Original and FDR-adjusted *p­*-values are now shown in the supplemental statistics tables.

After FDR of the pupillometry tests mentioned above, we see no significant difference between the unimodal and bimodal block sections of the visual rule. The difference that we previously noted between the unimodal and bimodal blocks within a rule, while consistent with our hypothesis and previous reports (Benjamini and Hochberg 1995), did not meaningfully contribute to the narrative. For the sake of brevity and consistent with Reviewer 3’s recommendation to simplify the figures, they have been moved to supplementary materials (Figure 2 —figure supplement 1).

5) Please also improve captions according to reviewer 2 recommendation. Simplifying the figures according to reviewer 3's recommendations (supplementary figures can be used) may help extend the captions without making them too long.

Each figure caption has been expanded to increase clarity and reduce dependence on text for understanding. Additionally, to streamline the figures, we have reduced the number of figure panels that provide redundant information or null results that do not contribute to the narrative. Such figure panels have been moved to the supplementary materials (Figure 2 —figure supplement 1 and Figure 7 —figure supplement 2.)

6) The discussion should be extended according to recommendations of reviewer 2 to include observations from previous studies and of reviewer 3 to include a better discussion of the mechanisms and biological meaning of the present findings (see public review). For the mechanisms, it should be particular attention should be put on the origin of suppression (inhibition or excitation)? The implication of attentional effects before stimulus presentation should also be clarified (referee 3).

We have expanded the Discussion section to address the points raised by the reviewers. In particular, we have included a section on the relevance of the work by Shamma, Fritz and colleagues on plasticity driven by attention in the auditory cortex (lines 447 to 464). Our results in Figure 6G suggest a sustained alteration of AC spectral representation, which does not appear to exhibit rapid spectral plasticity, at least when comparing A-rule and V-rule responses.

We have also addressed possible origins of attention-related suppression (Lines 486 to 496, beginning with “Previous studies have reported larger effects of task engagement or attention in inhibitory interneurons…”). To summarize, our division of units into NS and BS does not suggest a general increase in NS activity during auditory attention that would provide clear evidence of enhanced inhibitory network activity; instead, for both NS and BS units, modulation is heterogenous, but group level trends show that activity suppression is more common. However, this division of units by waveform only provides a coarse handle for excitatory/inhibitory network activity. Inhibitory subnetworks only identifiable through molecular tools rather than waveform classification may be driving suppression. We also cannot rule out decreased excitatory drive as a possible mechanism.

Finally, we have attempted to emphasize that attentional effects on pre-stimulus activity and task-irrelevant responses are likely to reflect modality-wide attentional effects that result from a reduction in spontaneous rates. This finding is therefore a central feature to our interpretation of the data. This is the primary message of Figure 5 (“Attention-related modulation of sound-evoked responses largely reflects pre-stimulus activity changes”). This is also consistent with previous work showing that reductions in spontaneous activity are a feature of attention or task engagement in multiple modalities and multiple model organisms (Buran, von Trapp, and Sanes 2014; Carcea, Insanally, and Froemke 2017 Yoshida and Katz 2011; Cox et al., 2019; Sato and Schall 2001; Bisley and Goldberg 2003; Herrington and Assad 2010). These citations are included in the Discussion (lines 435-440).

7) The authors should evaluate if suppression is more difficult to be picked up in supragranular layers, at the light of the repeated observation (also done by the group of Harris in the primary auditory cortex) that spiking frequencies are significantly lower (often reaching sparse levels, that is close to zero) there.

To address the possibility that low firing rates have prevented discovery of attentional effects in the superficial group, we identified (by randomly selecting from our recorded deep units, without replacement) a subset of deep units for which the distribution of firing rates matches the distribution of firing rates among the superficial units, and then performed the firing rate modulation analyses on this subset of deep cells. The deep group exhibited the greatest modulation by attentional state, so analyzing a subpopulation of these units with spike rates and sample size matched to superficial should tell us whether an effect can be observed from the population of superficial units we recorded. The results are presented in Author response image 2.

The top row shows the FR-matched subset of deep units (dark gray) selected to match the smaller population of superficial units (light gray). A_R_ and A_U_ responses are separated (L, R) for consistency with the paper. In the bottom row, modulation of the FR-matched subset is shown in colors (broad-spiking = red, narrow-spiking = blue), and is overlaid atop the modulation of the entire deep population (gray).In the FR-matched subset of deep units, we observe a significant effect of attention only for the A_R_* stimulus response. This shows that the sample size and firing rate distribution from the superficial group is sufficient to detect an effect in some but not all cases.

In our evaluation, this analysis suggests that it is possible to detect effects in populations of similar size and FR to our superficial group but does not rule out missed effects in superficial neurons, either due to biological or technical obstacles. We have added the following caveat to the discussion:

“One important caveat is that superficial AC is known to have lower spontaneous and evoked FRs than deeper cortex (Sakata and Harris 2009; Christianson, Sahani, and Linden 2011), which may have hindered the discovery of attention-related effects in our study. Furthermore, although we tried to minimize neural tissue damage through technical considerations such as using a slow probe insertion speed (Fiáth et al., 2019), the superficial layers likely sustain the greatest level of damage when the probe is inserted to span the full cortical depth. Despite these factors, we were able to isolate a reasonably large sample size of responsive neurons in superficial cortex from successful behavior sessions (n = 119 units, of which 57% were stimulus-responsive). Nevertheless, we cannot fully rule out that the absence of observed attentional modulation at superficial depths is not due to experimental limitations such as the comparatively small sample size. Future work employing imaging techniques to target superficial neurons may help resolve this.” Lines 473-485.

In addition to the compiled essential revisions, we have addressed the concern of Reviewer 2 regarding possible effects of low-level optogenetic pulses presented during inter-trial intervals. We have examined behavioral response error rate and stimulus response firing rates between trials immediately after optogenetic stimulation and stimulus-identity matched trials preceding the stimulation. No significant difference was observed in behavioral error rate post-opto vs. pre-opto: p = 0.21, Z = -1.27, error rates measured per recording, n = 21 recordings, Wilcoxon signed-rank test) or in response firing rate (p = 0.60, Z = 0.52,:

"In most experiments (n = 21 recordings), brief, low-level optogenetic pulses during the inter-trial interval of the task were used to identify opsin-expressing neurons (<0.3 mW light; 5 light pulses of 10 ms duration, every ~1.5 min); these analyses are outside of the scope of this report. The optogenetic stimulation protocol was consistent through A- and V-rules of the task. Unit stimulus response FRs and behavioral response error rates were similar between trials immediately after optogenetic pulses and stimulus-matched trials preceding the pulses.” Lines 530-536.

We again thank all of the reviewers for their thoughtful suggestions.***